# Uncovering an allosteric mode of action for a selective inhibitor of human Bloom syndrome protein

Xiangrong Chen[1,2], Yusuf I Ali[2,3], Charlotte EL Fisher[1], Raquel Arribas-Bosacoma[1], Mohan B Rajasekaran[3], Gareth Williams[3], Sarah Walker[3], Jessica R Booth[3], Jessica JR Hudson[3], S Mark Roe[4], Laurence H Pearl[1], Simon E Ward[3,5]*, Frances MG Pearl[2]*, Antony W Oliver[1]*

[1]Cancer Research UK DNA Repair Enzymes Group, Genome Damage and Stability Centre, School of Life Sciences, University of Sussex, Falmer, United Kingdom; [2]Bioinformatics Lab, School of Life Sciences, University of Sussex, Falmer, United Kingdom; [3]Sussex Drug Discovery Centre, School of Life Sciences, University of Sussex, Falmer, United Kingdom; [4]School of Life Sciences, University of Sussex, Falmer, United Kingdom; [5]Medicines Discovery Institute, Park Place, Cardiff University, Cardiff, United Kingdom

*For correspondence:
wards10@cardiff.ac.uk (SEW);
f.pearl@sussex.ac.uk (FMGP);
antony.oliver@sussex.ac.uk (AWO)

Competing interests: The authors declare that no competing interests exist.

**Abstract** BLM (Bloom syndrome protein) is a RECQ-family helicase involved in the dissolution of complex DNA structures and repair intermediates. Synthetic lethality analysis implicates BLM as a promising target in a range of cancers with defects in the DNA damage response; however, selective small molecule inhibitors of defined mechanism are currently lacking. Here, we identify and characterise a specific inhibitor of BLM's ATPase-coupled DNA helicase activity, by allosteric trapping of a DNA-bound translocation intermediate. Crystallographic structures of BLM-DNA-ADP-inhibitor complexes identify a hitherto unknown interdomain interface, whose opening and closing are integral to translocation of ssDNA, and which provides a highly selective pocket for drug discovery. Comparison with structures of other RECQ helicases provides a model for branch migration of Holliday junctions by BLM.

## Introduction

RECQ helicases catalyse the unwinding of duplex DNA with 3' to 5' directionality, driven by energy liberated by ATP-hydrolysis. As well as simple DNA duplexes, the various members of the RECQ helicase family (BLM, WRN, RECQ1, RECQ4, and RECQ5 in humans) are able to unwind DNA within a range of complex DNA structures and DNA repair intermediates, including: forks, bubbles, triple helices, displacement (D)-loops, G-quadraplexes, and three- or four-way Holliday junctions (extensively reviewed in *Croteau et al., 2014*; *Wu, 2012*).

RECQ-helicases are strongly implicated in the maintenance of genomic integrity, principally through their participation in the homologous recombination (HR) pathway for repair of DNA double-strand breaks and restart of collapsed or blocked replication forks (reviewed in *Croteau et al., 2014*; *Urban et al., 2017*), but also have roles in toleration of microsatellite instability (*Chan et al., 2019*; *Lieb et al., 2019*) and sister chromatid decatenation (*Chan et al., 2007*). Defects in RECQ-family members are responsible for rare genetic diseases displaying substantial genomic instability and cancer predisposition (*Bernstein et al., 2010*). Loss of function of WRN underlies the complex progeria Werner Syndrome; defects in BLM underlie Bloom Syndrome, which is characterised by growth retardation and immunodeficiency; and defects in RECQ4 are associated with Rothmund-

Thompson syndrome, which displays growth retardation, skeletal abnormalities and premature ageing.

A number of experimental and computational studies have implicated RECQ helicases – primarily BLM and WRN - as potential targets for cancer therapy, due to the synthetic lethality of their silencing or downregulation with genetic defects inherent in a range of different cancers (*Chan et al., 2019*; *Lieb et al., 2019*; *Aggarwal and Brosh, 2009*; *Behan et al., 2019*; *Datta et al., 2021*; *Kategaya et al., 2019*; *Pearl et al., 2015*; *Wang et al., 2018*). Despite the therapeutic opportunities this presents, no drugs targeting RECQ helicases have yet been licensed, although potential leads have been reported (*Nguyen et al., 2013*; *Rosenthal, 2010*; *Yin et al., 2019*).

Here, we determine the mode of action for two reported inhibitors of BLM – ML216 (*Nguyen et al., 2013*; *Rosenthal, 2010*) and a substituted benzamide (compound 2). While ML216 appears to act, at least in part, through direct DNA binding and has poor specificity, we find that 2 and derivatives thereof are highly specific binders of a defined BLM-DNA complex. Crystallographic analysis of the BLM-DNA-2 complex identifies a novel allosteric binding site and reveals a distinctive conformational step in the helicase mechanism, that can be trapped by small-molecules. These data pave the way for the development of allosteric inhibitors of BLM helicase with the potential to generate trapped and highly cytotoxic BLM-DNA complexes.

## Results

### Compound identification and screening

A series of compounds that targeted the helicase activity of human BLM were identified in a quantitative high-throughput screen (qHTS) (*Rosenthal, 2010*), where the results were made publicly available from the PubChem data repository [https://pubchem.ncbi.nlm.nih.gov/bioassay/2528]. Filtering the 627 reported active compounds for preferential physicochemical properties (e.g. Lipinski's rule of five) and excluding those with potential pan-assay interference activity (PAINS) allowed us to group the compounds into several distinct clusters according to chemical similarity. The inhibitory activity of exemplars from each cluster were tested in a fluorescence-based DNA unwinding assay (*Rosenthal, 2010*) against recombinant human BLM-HD (HD = helicase domain; amino acids 636–1298). However, only a single compound produced an $IC_{50}$ lower than 10 μM (compound 1, $IC_{50}$ = 4.0 μM; *Figure 1A*).

We synthesised and purified six close analogues of this compound with the aim of generating preliminary structure-activity relationship data and confirmed their inhibitory activity in the unwinding assay (Materials and methods, Appendix 1). Compounds 2 to 6 inhibited the 3' → 5' helicase activity of recombinant human BLM-HD with $IC_{50}$ values ranging from 2.2 to ~60 μM, whereas 7 did not inhibit BLM-HD over the concentration range tested (*Figure 1A* and *Figure 1—figure supplement 1*, *Table 1*). An $IC_{50}$ of 4 μM was determined for ML216, a compound reported to be a semi-selective inhibitor of human BLM (*Nguyen et al., 2013*; *Rosenthal, 2010*), which was included as a positive control (*Figure 1A*).

In a malachite green-based assay that measures ATP turnover, we observed robust stimulation of hydrolysis by BLM-HD when the protein was incubated with a short single-stranded 20-base oligonucleotide (*Figure 1B*). Here, we determined $IC_{50}$ values ranging from 3.2 to ~50 μM for each of our active analogues and 4.4 μM for ML216 (*Figure 1C* and *Figure 1—figure supplement 1*, *Table 1*). Whilst the values of $IC_{50}$ obtained in our orthogonal assay did not agree in absolute value with those determined in the first, it ranked each analogue with a similar order of potency.

### Biophysical analysis of compound binding

We could readily observe changes in fluorescence, indicative of binding, upon titration of both ADP and ATP-γS into BLM-HD using microscale thermophoresis (MST, *Figure 1—figure supplement 2*). We could not, however, observe any interaction for our most potent compound 2. In the absence of biophysical evidence for binding, we sought to confirm that 2 wasn't just a false positive generated by interference with the fluorescent readout of the unwinding assay. An alternative gel-based assay allowed direct visualisation of the conversion of a forked DNA-duplex into its component single-stranded oligonucleotides via the helicase activity of BLM-HD (*Figure 2A*). Titration of 2 clearly

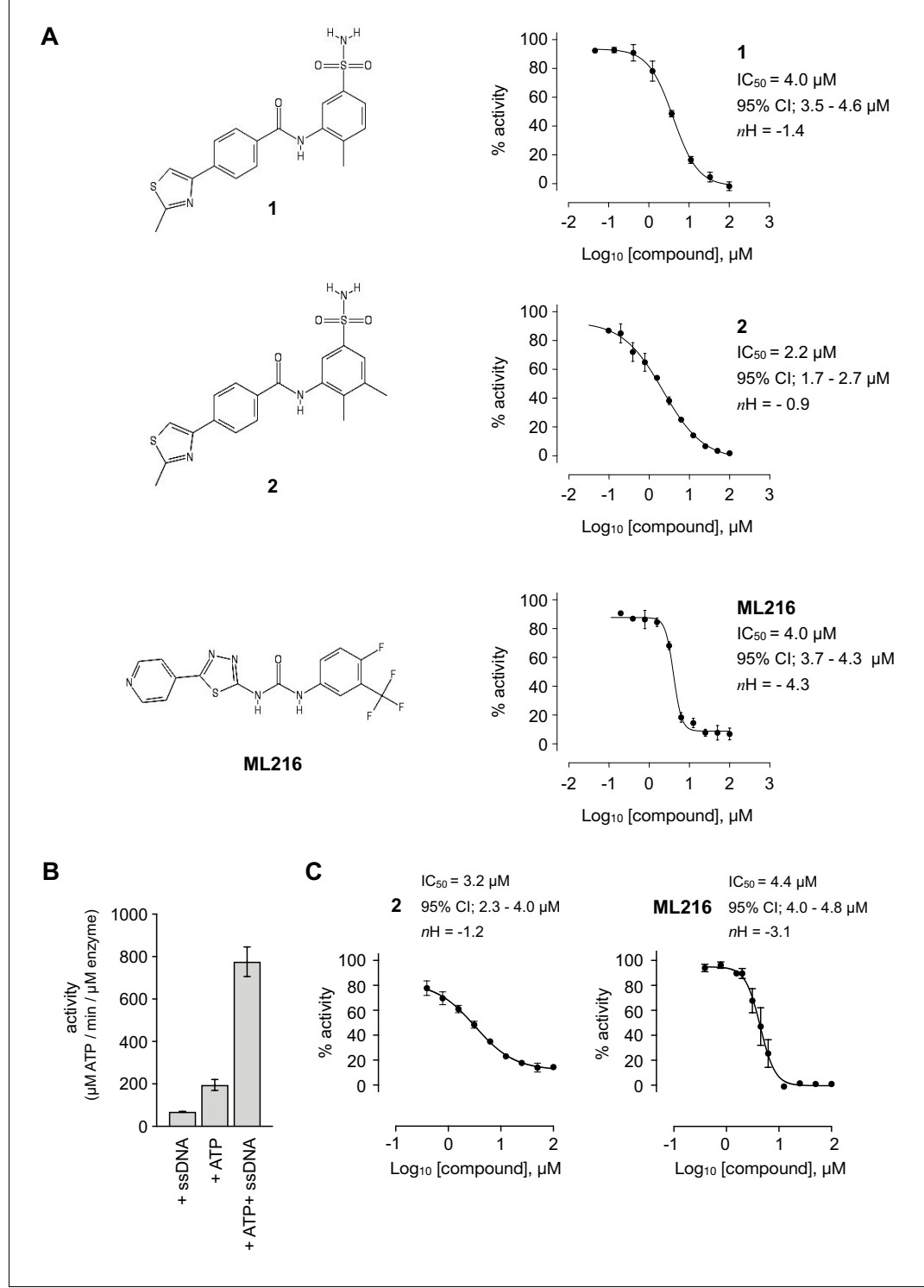

**Figure 1.** Inhibition of BLM helicase unwinding activity. (**A**, left) Chemical drawings for compounds 1, 2, and ML216. (**A**, right) Dose response curves from fluorescence-based DNA unwinding assays with BLM-HD. Experimental data were fitted with a four parameter, log(inhibitor) vs. response model with variable slope. Calculated values for $IC_{50}$, Hill slope ($nH$) and 95% confidence intervals (95% CI) are given in each case. (**B**) Turnover of ATP by BLM-HD, as measured by a malachite-green end-point assay, is strongly stimulated in the presence of a 20-base single-stranded oligonucleotide. (**C**) Dose response curves from ATP-turnover assays with BLM-HD. Data were fitted as for (**A**). In each case data points are the mean of three technical replicates, with error bars representing one standard deviation (1 SD).

*Figure 1 continued on next page*

*Figure 1 continued*

The online version of this article includes the following figure supplement(s) for figure 1:

**Figure supplement 1.** Assay data for compunds 3 to 7.

**Figure supplement 2.** Indicative binding isotherms for titrations of Mg-ADP and Mg-ATPγS into BLM-HD as measured by MST.

inhibited production of the single-stranded DNA product in a dose-dependent manner, with a calculated $IC_{50}$ of 1.8 µM (*Figure 2B*).

## ML216 interacts with DNA

Another potential false positive could be generated by compounds that bind directly to DNA, thus preventing BLM-HD from productively engaging with its substrate. To test this hypothesis, we used a commercial assay that utilises recombinant Topoisomerase I (Topo I) to relax a supercoiled plasmid. Compounds that intercalate or bind to the major or minor groove of the plasmid DNA prevent relaxation. At the manufacturer's recommended concentration of 200 µM, the positive control m-Amsacrine (mAMSA) strongly inhibited relaxation of the supercoiled plasmid. In contrast, no effect was observed with 2 at the same concentration. However, partial inhibition of relaxation could be observed for a reaction containing ML216 (*Figure 2C*). To confirm this observation, we purchased ML216 from an alternative commercial supplier (ML216-A) and also resynthesised and purified the compound in-house (ML216-B; Materials and methods, Appendix 1). In both cases, a similar level of inhibition was observed when the compounds were included in the relaxation assay, indicating that this was both a real and reproducible effect (*Figure 2C*).

To explore further the possibility that ML216 might interact directly with DNA, we tested its ability to displace SYBR Green II (SG2) from a DNA substrate in a dye displacement assay (*Del Villar-Guerra et al., 2018*; *Tse and Boger, 2004*). When SG2 binds to DNA, a concomitant increase in its fluorescence can be measured. If an added compound can compete with the dye for binding to the DNA, a corresponding decrease in the fluorescent signal is observed. We titrated ML216 into a forked-50mer dsDNA substrate, that had been pre-incubated with SG2, observing a clear time- and dose-dependent displacement of the dye, indicating that ML216 can directly interact with a DNA substrate (*Figure 2D*).

## Compound 2 does not interfere with ATP-binding

With confidence that 2 was, in fact, a bona fide inhibitor of BLM, we repeated the unwinding assay in the presence a 10-fold higher concentration of ATP to examine if the compound was directly competitive with nucleotide binding to the active site of the enzyme. As the resulting $IC_{50}$ value was identical to that previously determined, it ruled out this mode of inhibition, and suggested that the compound bound elsewhere (*Figure 2—figure supplement 1*).

## Compound 2 is a non-competitive inhibitor

ATP-turnover experiments, under Micheation-Menten conditions, allowed us to generate a Lineweaver-Burk plot with data taken from DNA substrate titrations in the presence of 0, 5, and 10 µM of 2. The resultant plot indicated a non-competitive (allosteric) mode of inhibition for 2 (*Figure 2E*). With this information, we postulated that 2 might only bind to BLM-HD when it was engaged with a DNA substrate. We therefore revisited MST, first confirming the interaction of BLM-HD with the single-stranded 20mer used in our malachite green assay, plus a shorter 15mer that would be taken into crystallographic trials (*Figure 2F*). We next titrated 2 into the two pre-formed BLM-HD/ssDNA complexes. This time changes in fluorescent signal could be detected, confirming our hypothesis, with dissociation constants of 1.7 and 2.6 µM determined for the interaction with the 15mer and 20mer, respectively (*Figure 2G*).

## Enabling structural biology with the expression construct BLM-HD$^{\Delta WHD}$

We created the expression construct BLM-HD$^{\Delta WHD}$ to remove the conformationally flexible Winged Helix domain (WH) that requires the presence of either a stabilising nanobody, or interaction with a large DNA substrate to facilitate crystallogenesis (*Newman et al., 2015*; *Swan et al., 2014*)

**Table 1.** Summary of inhibition data for seven exemplars from the identified compound series. $IC_{50}$ values were determined by fitting of experimental data to log (inhibitor) vs response models provided in GraphPad Prism. Data for the unwinding assay correspond to three technical replicates from a single experiment. For the ATP-turnover assay data correspond to at least two independent experiments, each containing three technical replicates.

| # | Chemical drawing | unwinding $IC_{50}$ [95% CI]; µM | turnover $IC_{50}$ [95% CI]; µM |
|---|---|---|---|
| 2 | | 2.2 [1.7–2.7] | 3.2 [2.3–4.0] |
| 3 | | 3.5 [2.4–5.2] | 5.3 [4.7–6.1] |
| 4 | | 6.6 [3.4–12.7] | 11.2 [8.3–15.3] |
| 5 | | 12.8 [4.5–36.8] | 47.86 [18.28–180.7] |
| 6 | | 56.9 [25.4–171.3] | 40.94 [15.9–139.8] |
| 7 | | No inhibition | No inhibition |
| ML216 | | 4.0 [3.7–4.3] | 4.4 [4.0–4.8] |

replacing it with a short poly-(glycine/serine) linker that serves to connect the Zinc-binding domain (Zinc) directly to the Helicase and RNAse C-terminal domain (HRDC, *Figure 3A*). In validation of this approach, we were able to crystallise the protein in complex with ADP and magnesium co-factor, and to determine its structure at a resolution of 1.53 Å; a significant increase in resolution over structures previously deposited in the PDB (4CDG, 2.8 Å; 4CGZ, 3.2 Å; 4O3M, 2.3 Å; see *Appendix 1—table 1*).

Superposition of the structures of BLM-HD (PDB: 4CDG) and BLM-HD$^{\Delta WHD}$ produces a rmsd of 0.86 Å over 2450 atom positions (D1 + D2 + Zn; PyMOL), indicating the overall conformation and geometry of the two recombinant proteins is highly similar, despite deletion of the WH domain (*Figure 3—figure supplement 1*). Furthermore, BLM-HD$^{\Delta WHD}$ binds both ssDNA-15mer and 2 with a similar affinity to that of BLM-HD (*Figure 3—figure supplement 2*).

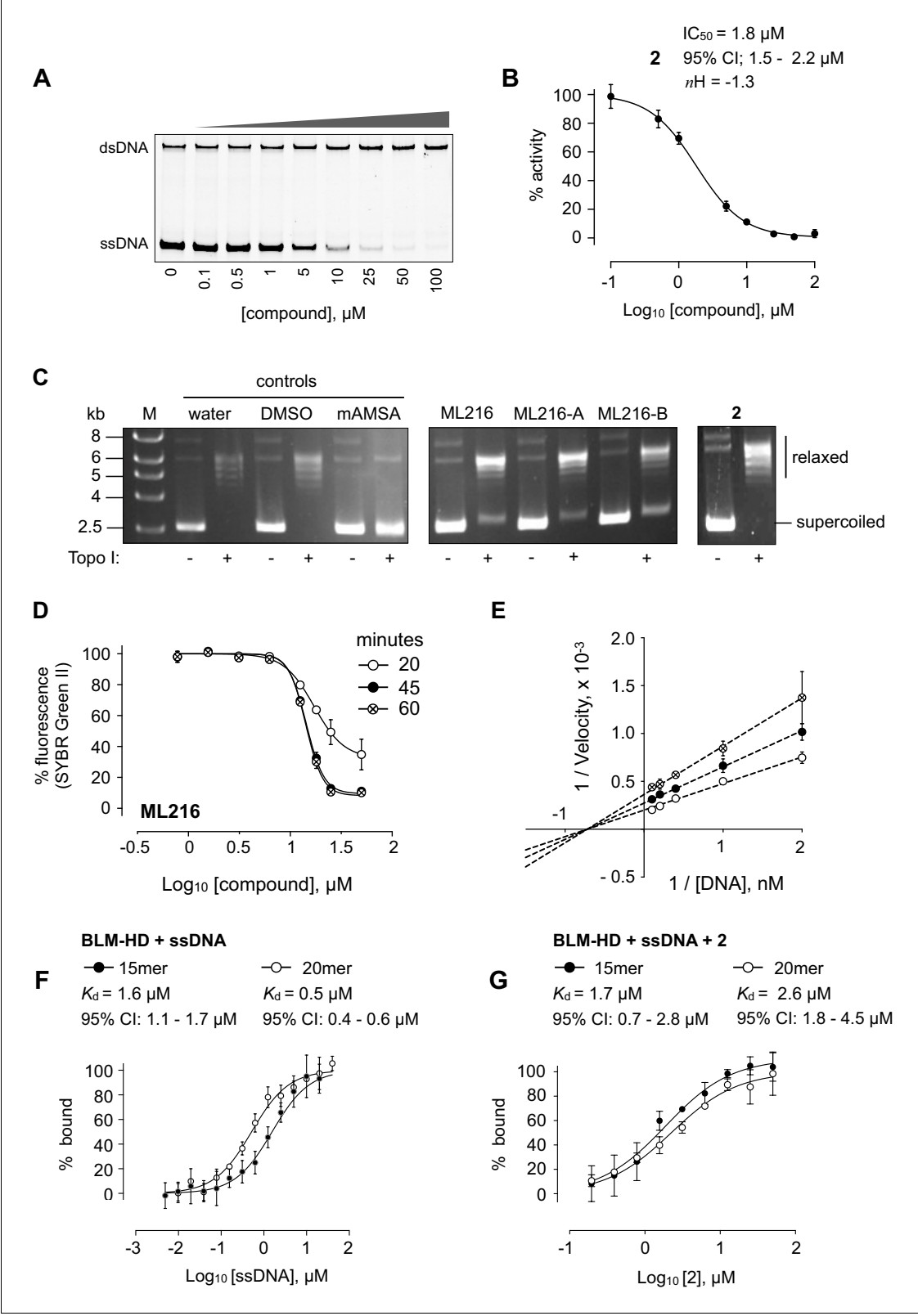

**Figure 2.** DNA interaction assays. (**A**) Titration of BLM-HD with 2 prevents the unwinding of a forked-50mer dsDNA substrate into its component strands, as judged by native gel electrophoresis. (**B**) Quantification of inhibitory activity by 2 in the gel-based activity assay. Experimental data were fitted with a four parameter, log(inhibitor) vs. response model with variable slope. Calculated values for IC$_{50}$, $n$H and 95% CI are given in each case. (**C**) Representative results from a Topoisomerase I (Topo I) DNA-unwinding assay. M = molecular mass maker; DMSO = buffer supplemented with dimethyl

*Figure 2 continued on next page*

*Figure 2 continued*

sulfoxide control; mAMSA = mAmsacrine; ML216, ML216-A, ML216-B = refer to the three independent sources of the compound as described in the main text of the manuscript (D) Dose response curves from SYBR-Green II dye displacement assays, using a forked-50mer DNA duplex incubated with ML216 for a period of 20 (open circles), 45 (filled circles), and 60 min (crossed circles). Fitted lines are intended as visual aids only. (E) Lineweaver-Burk plot for data generated at three compound concentrations (0, 5, and 10 µM) in a colourimetric ATP turnover assay. Linear regression produces an intercept of all data on the X-axis indicating that 2 is a non-competitive inhibitor (i.e. same $K_m$, altered $V_{max}$ parameter). (F, G) Binding isotherms for binding of BLM-HD to ssDNA-15mer and −20mer, or to compound 2 in the presence of either oligonucleotide, as determined by microscale thermophoresis (MST). Experimental data were fitted with a one-site, specific binding model. Values for $K_d$ and 95% CI are given in each case. For all plots, data represent the mean of three technical replicates with error bars representing 1 SD.

The online version of this article includes the following figure supplement(s) for figure 2:

**Figure supplement 1.** Dose response curves from fluorescence-based DNA unwinding assays with BLM-HD, carried out at two different ATP concentrations.

## Crystal structure of BLM-HD$^{\Delta WHD}$ in complex with compound 2 and ssDNA

We crystallised 2 in complex with BLM-HD$^{\Delta WHD}$, ADP/magnesium co-factor, and ssDNA-15mer (liganded complex); determining its structure at a resolution of 3.0 Å (*Appendix 1—table 1*). The complex crystallised in space group P1, with six molecules of BLM-HD$^{\Delta WHD}$ and associated ligands forming the asymmetric unit. Interestingly, the co-crystallised ssDNA-15mer helped drive formation of the crystal lattice, due to its partial self-complementary at the 5′ end (5′ –CGTAC–3′) that serves to form four consecutive base pairs between two oligonucleotides (*Figure 3B*); the cytosine at the 5′ end of the oligonucleotide is not readily discernible in electron density maps and is therefore likely to be disordered. An extensive series of interactions are made to the bound nucleic acid by amino acids from all four sub-domains of the BLM-HD$^{\Delta WHD}$ expression construct (*Figure 3—figure supplement 3*).

Compound 2 sits in a small pocket found on the opposite face of the protein to that which binds nucleotide (*Figure 3B*, inset), and integrates amino acid side chains from both the D1 and D2 subdomains of the helicase core, as well as several from the Zn-binding domain. The oxygen of the amino group at the centre of the compound is hydrogen-bonded to the side chain of Asn1022, whilst the nitrogen of the same moiety is in hydrogen-bonding distance to both the backbone oxygen and side chain of Ser801. The side chains of His805 and Thr1018 stack up against, and provide Van der Waals contacts to, the central ring system of the 3-amino-4,5-dimethylbenzenesulfonamide pendant group as part of a pocket lined by the side chains of residues Asp806, His1014, Thr1015, His1019 (*Figure 3C*). The nitrogen of the sulphonamide group is hydrogen-bonded to the side chain of Asp840, which itself is bonded to the side chain of His805. The 2-methyl-thiazole moiety of 2 sits against the surface of the alpha-helix containing Gly972 and is sandwiched by additional packing interactions with the side chains of Gln802 and Glu971. The side chains Thr832 and His798 also contribute to this section of the binding pocket, which is 'capped' by Gln975. The central benzene ring of 2 is also contacted by the side chains of Gln802 and Glu971.

## Reconfiguration of the aromatic-rich loop

The aromatic-rich loop (ARL) is a highly conserved motif in RecQ helicases that serves as a molecular 'sensor', detecting binding of single-stranded DNA and coupling it to structural rearrangements that enable ATP hydrolysis (*Manthei et al., 2015*; *Zittel and Keck, 2005*). In our high-resolution structure of BLM-HD$^{\Delta WHD}$, the ARL is disordered and is not visible in electron-density maps (*Figure 4A*). By contrast, it can be fully modelled in the liganded complex, but its conformation is distinct from that observed in PDB entries 4CDG and 4CGZ where the single-stranded extension of bound nucleic acid substrates does not extend across to the D1 domain (*Figure 4B*, *Figure 4—figure supplement 1*).

Structures with a high degree of structural similarity to the liganded complex were identified with PDBeFold (*Krissinel and Henrick, 2004*). The search produced PDB entries 6CRM and 4TMU, with Q-scores of 0.47 and 0.46 respectively, which both describe structures of the catalytic core of *Cronobacter sakazakii* RecQ helicase (CsRecQ) in complex with different DNA substrates (*Manthei et al., 2015*; *Voter et al., 2018*). Comparison, in each case, reveals a close to identical conformation of the

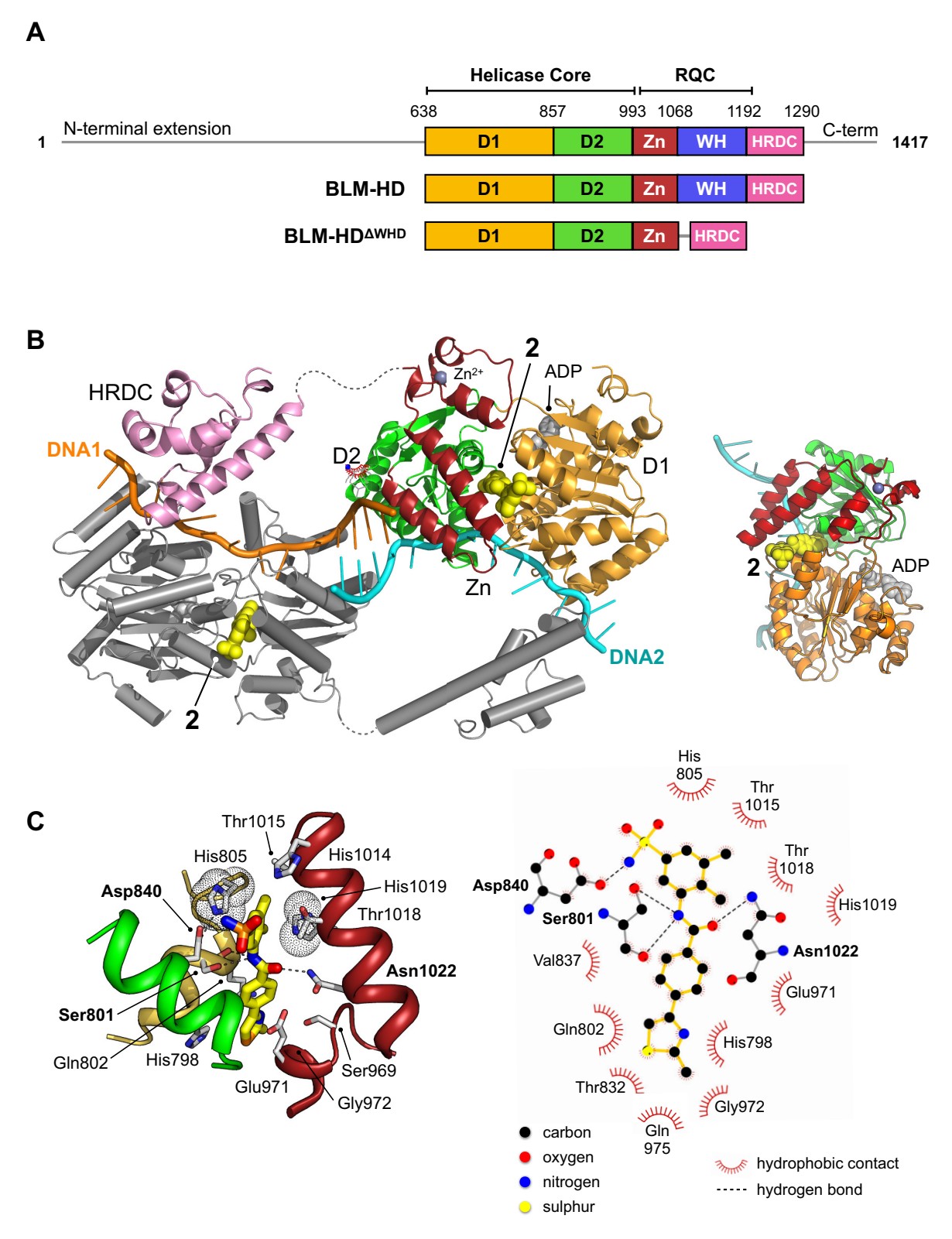

**Figure 3.** Structural overview of BLM-HD^ΔWHD in complex with compound 2. (**A**) Schematic representation of the domain composition and respective amino acid boundaries for full-length human BLM and the two expression constructs used in this study BLM-HD and BLM-HD^ΔWHD. D1 and D2 = domains 1 and 2 of the helicase core; Zn = zinc-binding domain; WH = winged-helix; HRDC = Helicase and RNaseD C-terminal domain; RQC = RecQ C-terminal domain. (**B** and **B** inset) Molecular secondary structure cartoon highlighting components of a 'pseudo-symmetrical' dimer found in the

*Figure 3 continued on next page*

*Figure 3 continued*

asymmetric unit of BLM-HD$^{\Delta WHD}$/Mg-ADP/**2**/ssDNA crystals, driven by partial complementarity of the single-stranded 15mer oligonucleotide at its 5' end (DNA1 and DNA2, coloured orange and cyan respectively). Compound 2 (yellow-coloured spheres) binds to a small pocket found on the opposite side to that which binds nucleotide (grey-coloured spheres). (C, left) Molecular cartoon representation highlighting interactions made between 2 and BLM-HD$^{\Delta WHD}$. Key amino acid residues are labelled and shown in stick representation, with carbon atoms coloured according to the schematic shown in panel A. Compound 2 is shown in stick representation, with carbon atoms coloured yellow. Potential hydrogen bonds are indicated by black dotted lines. (C, right) Modified LIGPLOT+ (*Laskowski and Swindells, 2011*) diagram of protein-compound interactions. See associated key for additional detail.

The online version of this article includes the following figure supplement(s) for figure 3:

**Figure supplement 1.** Superposition of the structures of BLM-HD (PDB: 4CDG) and BLM-HD$^{\Delta WHD}$ using PyMOL (*Schrödinger, 2020*).

**Figure supplement 2.** Isotherms for binding of BLM-HD$^{\Delta WHD}$ to (top) ssDNA-15mer and (bottom) compound 2 in the presence of ssDNA-15mer, as determined by MST.

**Figure supplement 3.** Schematic summary of DNA-interactions made within the crystal lattice of the liganded complex (BLM-HD$^{\Delta WHD}$ + Mg-ADP + ssDNA-15mer + compound 2).

ARL to that observed in our liganded complex, as well as nucleic acid interactions that include the D1 domain (*Figure 4—figure supplements 1*, *2* and *3*).

For CsRecQ, *Manthei et al., 2015* described concerted movements of residues Phe158 and Arg159, within the ARL to interact with the 3' single-stranded extension of their co-crystallised DNA substrate in our liganded complex the equivalent residues undergo a similar transition (Phe807 and Arg808, respectively). We observe that Phe158 moves to make base-stacking interactions with G8 and A9 of the bound ssDNA-15mer. By comparison to 4CGZ, we also see that Arg808 switches from interacting with Asp806 of the ARL to the backbone oxygen of Pro715 and the side chain of Glu768 (*Figure 4A and B*). Notably, mutation of residues equivalent to Arg808 or Glu768 in EcRecQ (Arg159 and Glu124) have been shown to perturb enzyme function (*Manthei et al., 2015*; *Zittel and Keck, 2005*). Interestingly, in BLM, Asp806 is 'freed' to interact with the side chain of His1019, a residue within the Zn-binding domain (*Figure 4A and B*).

## Movement of the HRDC from parked to DNA-engagement

The HRDC (*H*elicase and *R*Nase *D* C-terminal domain) was originally identified as a putative nucleic-acid binding motif in both BLM and WRN (Werner syndrome helicase) and named in part for its similarity to a domain found at the C-terminus of *E. coli* RNase D (*Morozov et al., 1997*). However, only very weak ssDNA binding ($K_d$ ~100 µM) has been reported for this domain in isolation (*Kim and Choi, 2010*).

In their paper describing the crystal structure of BLM in complex with DNA, Newman et al. observe that the HRDC domain packs against a shallow cleft formed between the D1 and D2 domains of the helicase core, with the interface between the different modules being highly polar in nature. Their follow-on small angle X-ray scattering experiments also indicate that the HRDC domain of BLM is free to disassociate and re-bind to the helicase core (*Newman et al., 2015*). In our high-resolution crystal structure of BLM-HD$^{\Delta WHD}$, we observe the same 'parked' interaction for the HRDC, even in the absence of the WHD, but in our liganded structure we see that the HRDC domain swings across the face of the core helicase fold (*Figure 5A*) to make a series of polar interactions with the ssDNA-15mer, which is presented on the surface of the D1 domain of a second protomer within the asymmetric unit (*Figures 5B* and *3B*). The side chains of HRDC residues Asn1242 and His1236, contained within the 'hydrophobic $3_{10}$ helix' (*Kim and Choi, 2010*), are hydrogen-bonded to the O4 group of the T12 base and the N3 of the G11 base, respectively. The side chain of Phe1238 is also involved in an edge to ring-stacking interaction with the G11 base.

Each of these interactions is consistent with chemical shift changes previously observed in HSQC spectra – as a result of titrating ssDNA into $^{15}$N-labelled BLM-HRDC (*Kim and Choi, 2010*) – suggesting that the observed interactions have biological relevance, and that our structure represents the first to capture HRDC interactions with ssDNA. Furthermore, amino acids residues Lys1227, Tyr1237, Thr1243, and Asn1239 are also in close proximity to the bound DNA (*Figure 5B*) and could be expected to undergo changes in chemical environment upon interaction; again consistent with the reported perturbations in HSQC spectra (*Kim and Choi, 2010*). Asn1239 might also be expected

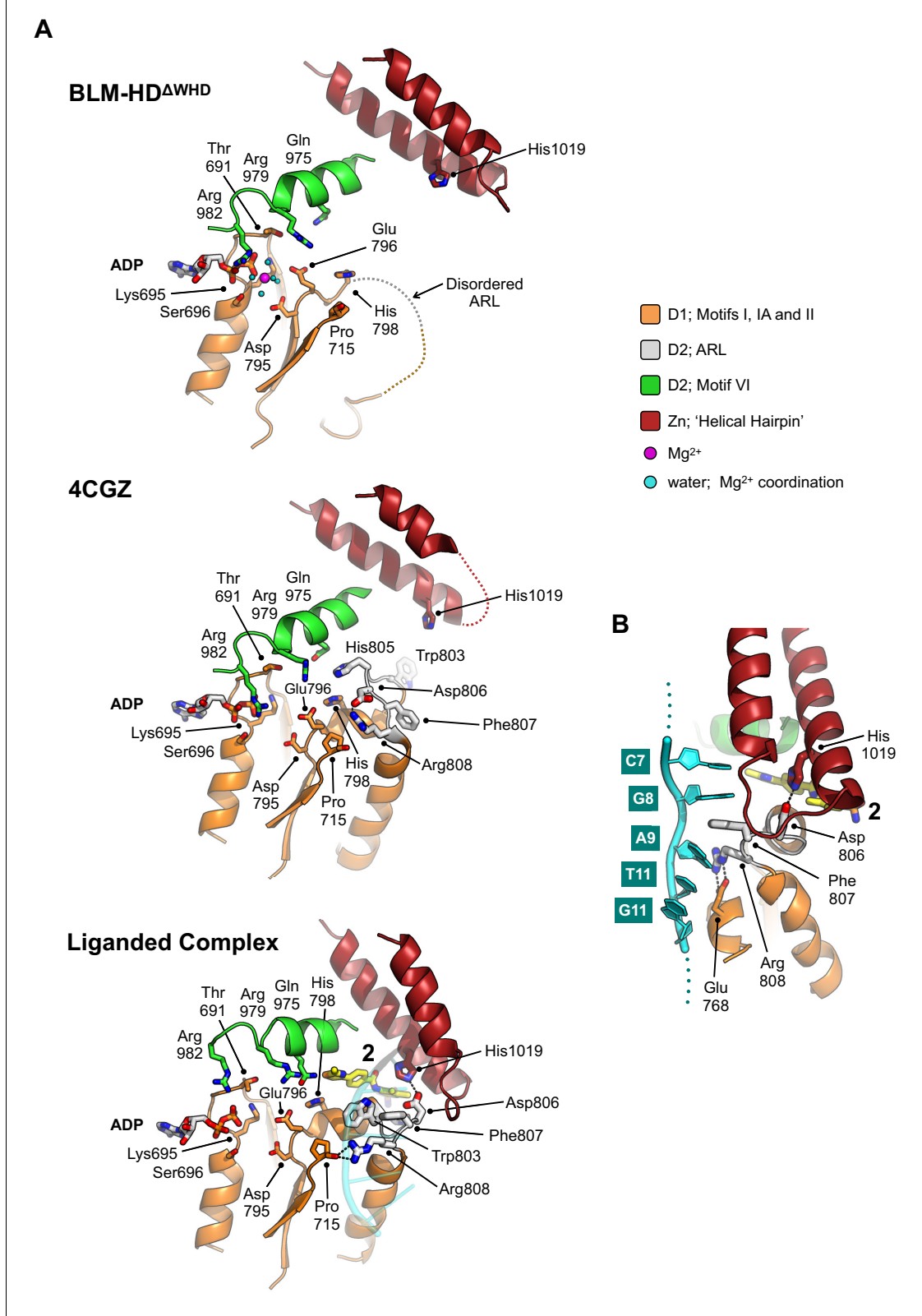

**Figure 4.** Structural transitions around the aromatic rich loop. (**A**) Molecular secondary structure cartoons for the region surrounding the aromatic rich loop (ARL) of BLM-HD$^{\Delta WHD}$ (top), PDB entry 4CGZ; BLM-HD in complex with DNA (middle) and liganded complex; BLM-HD$^{\Delta WHD}$ in complex with ADP, ssDNA-15mer and 2 (bottom). The side chains for key amino acid residues are shown in stick representation, with carbon atoms coloured according to their respective domains (see associated key). Bound ADP and 2 are also shown in stick representation, with carbon atoms coloured grey and yellow,

*Figure 4 continued on next page*

*Figure 4 continued*

respectively. (B) Expanded and rotated view highlighting the interactions made between the ARL and ssDNA-15mer oligonucleotide (cartoon coloured cyan) in the liganded complex, also showing the relative position of compound 2. Potential hydrogens bonds are represented by black dotted lines.

The online version of this article includes the following figure supplement(s) for figure 4:

**Figure supplement 1.** Molecular secondary structure cartoons showing selected amino acid side chains of the aromatic-rich loop (ARL) region in PDB entries 4CDG and 4CGZ (BLM-HD) and 4TMU (*Cronobacter sakazakii* RecQ) to that reported here for liganded-BLM-HD$^{\Delta WHD}$.

**Figure supplement 2.** Superposition of the ARL in liganded-BLM-HD$^{\Delta WHD}$ (coloured orange) with those found in PDB entries 4TMU (cyan) and 6CRM (*Voter et al., 2018*) (grey); which both represent structures of the catalytic core of *C. sakazakii* RecQ in complex with different DNA substrates.

**Figure supplement 3.** Molecular cartoon representations of the helicase catalytic cores reported in PDB entries 4CGZ, 4O3M, and 4TMU, highlighting their respective interactions with bound DNA substrates and comparing this to liganded BLM-HD$^{\Delta WHD}$.

to pick up an additional contact with the 5'-phosphate of a subsequent nucleotide in an extended substrate.

## Examining the selectivity profile of compound 2

With a robust molecular understanding for the binding mode for 2, we next examined if the compound displayed selectivity for members of the RecQ-helicase family. In our ATP-turnover assay, we saw no inhibition of recombinant helicase domains (HD) from human WRN, human RecQ5 or the unrelated *E. coli* helicase UvrD over the concentration range tested, whilst at higher concentrations inhibitory effects started to appear against human RecQ1. In contrast, ML216 robustly inhibited all four RecQ-family helicases tested and at higher concentrations also affected UvrD; in line with, and in support of, our observation that ML216 is non-specific and elicits at least part of its inhibitory effect by binding directly to DNA (*Figure 6A*).

Whilst compound solubility prevented generation of a complete inhibition curve for UvrD and thus a robust estimate of IC$_{50}$, the estimated Hill coefficient (*n*H) was close to 1 — in contrast to those calculated for titrations of ML216 against the RecQ helicases, which were generally steeper (ranging from 1.8 to 4.3), again suggesting that instead of a forming a 1:1 protein to inhibitor complex, there is in fact, a more complex (possibly mixed) mode of binding for this compound to this class of enzymes.

## Conformational trapping by compound 2

As binding of 2 has no direct effect on the ability of BLM-HD$^{\Delta WHD}$ to bind either ssDNA or nucleotide, we hypothesised that it might act to 'lock' the helicase into a conformational state where DNA substrates remain bound but cannot be unwound. In support of this idea, we undertook MST assays with a labelled single-stranded oligonucleotide in both the presence and absence of 2. Here, we observed a clear concentration-dependent decrease in $K_d$ when 2 was added, consistent with a decrease in the off-rate for DNA-binding, supporting our hypothesis that the compound acts to 'trap' BLM in its interaction with ssDNA (*Figure 6B*).

## Discussion

### Conformational cycle of RecQ helicases

During their catalytic cycle, the RecQ-family of helicases undergo a sequential set of conformational transitions, driven by ATP-binding and hydrolysis, then the subsequent release of inorganic phosphate and ADP (*Newman et al., 2017*). An initial 'encounter' complex is generated when the D2 domain of a RecQ-helicase binds to single-stranded DNA (ssDNA). This then leads to a set of motif / domain movements, including the ARL, that serve to couple ATP-binding and hydrolysis to movement of the bound ssDNA, such that it is now functionally engaged with the D1 domain. The process of DNA-unwinding is thought to proceed via an 'inchworm' type of mechanism, where the D2 domain sequentially binds and releases the DNA substrate, in order to 'feed' it onto the D1 domain, translocating it one base at a time. The WHD and HRDC domains of the RecQ helicase-family appear to contribute to the binding, recognition and unwinding of different DNA substrates via their ability to bind to ds and ssDNA respectively.

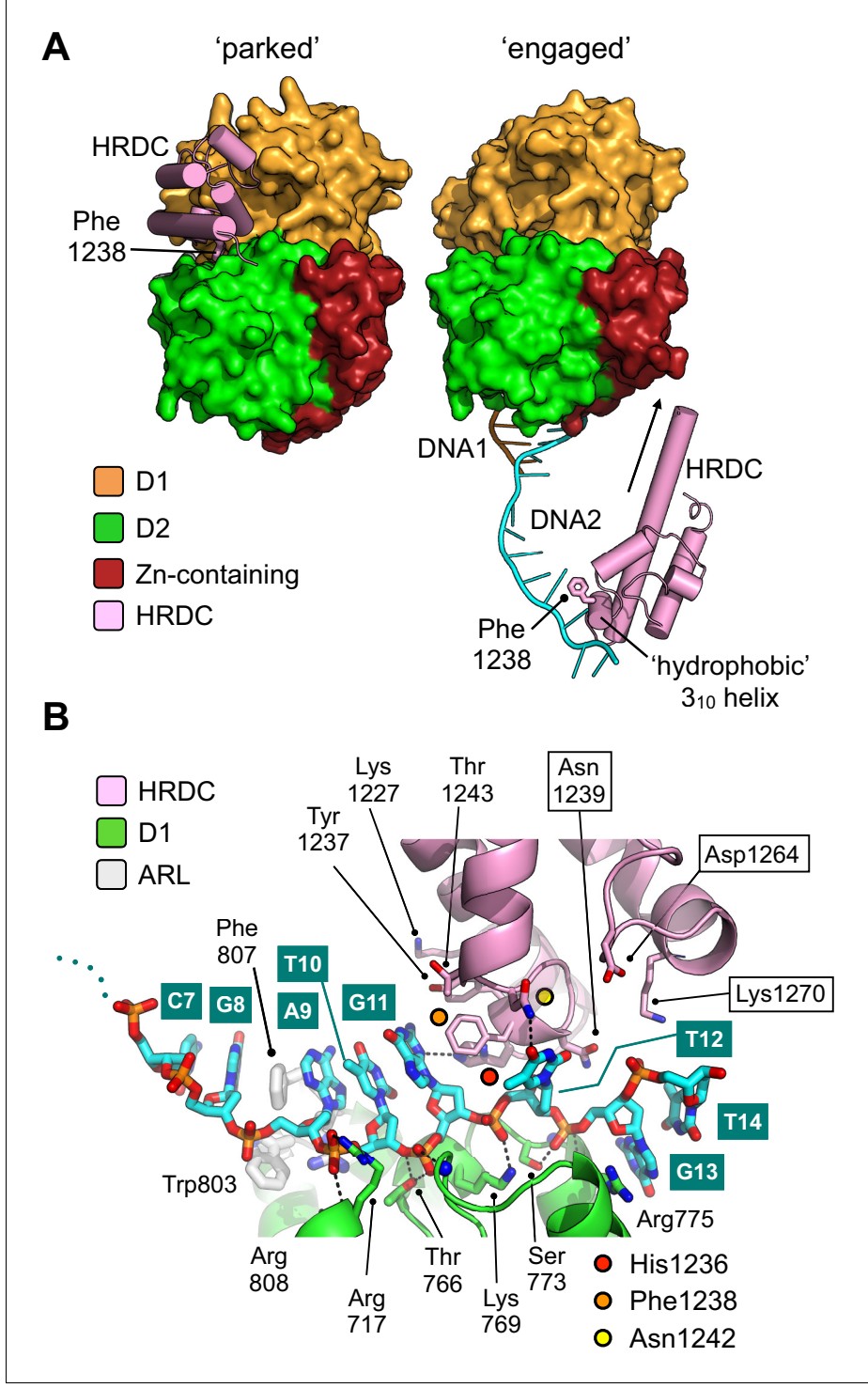

**Figure 5.** Repositioning of the HRDC domain. (A) Molecular surface representation of BLM-HD$^{\Delta WHD}$ (left) and the liganded complex (right) highlighting the relative positions of the HRDC domain (cylindrical helices coloured in pink). The HRDC moves from a 'parked' position located on one side of the helicase core, to an 'engaged' position on the other side in order to interact with the bound ssDNA-15mer. The N-terminus of the first HRDC alpha-helix is extended at the by ~6 aa, relative to the 'parked' position (as indicated by an arrow). (B) Molecular secondary structure cartoon highlighting interactions made by the HRDC to the bound ssDNA-15mer oligonucleotide. Side chains for key amino acid residues are shown in stick representation, with carbon atoms coloured according to their respective domains (see associated key). The bound ssDNA-15mer oligonucleotide is

*Figure 5 continued on next page*

*Figure 5 continued*

involved in interactions with both the D1 domain (carbon atoms coloured green) and the HRDC (carbon atoms coloured pink). Bound oligonucleotide is shown in stick representation, with carbon atoms coloured cyan.

## Inhibition via the allosteric binding site

As noted earlier, our liganded complex is structurally most similar to PDB entry 4TMU (CsRecQ)—a structure that has been previously characterised as representing a 'pre-ATP hydrolysis' conformation in the catalytic cycle of RecQ-family helicases (*Newman et al., 2017*). Although we have co-crystal-lised BLM-HD$^{\Delta WHD}$ with ADP, we can see that the active site of the liganded complex is also compatible with ATP-binding, through comparison to the crystal structure of *E. coli* RecQ in complex with ATP-γ-S (*Figure 6—figure supplement 1*; *Bernstein et al., 2003*). This in turn suggests that binding of compound 2—to a small pocket hereinafter referred to as the allosteric binding site (ABS)—acts to 'trap' or stall BLM by blocking the set of conformational changes required for progression to the next step of the catalytic cycle.

The ARL must be in a particular conformation to permit compound binding, otherwise the side chain of Trp803 would occupy and occlude the ABS (*Figure 4A*, *Figure 4—figure supplement 1*). This condition is satisfied when the D1 domain of BLM is engaged with single-stranded DNA, providing one explanation as to why we only observe compound binding in the presence of oligonucleotide. However, it is equally possible that binding of 2 serves to promote rearrangement of the ARL into the configuration that then permits the D1-ssDNA interaction.

The side chains of amino acids Gln975 and His798 contribute to the ABS (*Figure 3C* right and *Figure 4A*). In hsRECQ5, a polar contact between the equivalent residues (Gln345 and His160) is reported to be broken as a consequence of ATP-binding, freeing the ARL to interact productively with the single-stranded 3'-overhang of a bound DNA substrate (*Newman et al., 2017*). Importantly, mutation of the glutamine to alanine (Q345A) prevents stimulation of ATPase activity by binding to DNA, but does not perturb the basal rate of hydrolysis (*Newman et al., 2017*). The importance of the conserved glutamine is perhaps more apparent in *E. coli* RecQ, where mutation of the equivalent residue (Q322A) essentially ablates all ATPase capability (*Manthei et al., 2015*). Consistent with the 'pre-ATP hydrolysis' conformation for our liganded complex, the polar contact between Gln975 and His798 is broken, being ~3.8 Å apart. We note, however, that proximity of the 2-methyl-thiazole moiety from 2, may sterically prevent reformation of this important contact.

It is clear that more detailed kinetic studies would be required to unambiguously distinguish between a 'passive' or 'induced' mode of binding for 2. However, compounds bound stably to the ABS should sterically prevent the ARL from reverting to its initial conformation/structurally disordered state found at the beginning of the catalytic cycle (*Figure 4A,B*). The observed hydrogen bond between Asp806 and His1019 may also act to stabilise the ternary interaction between BLM-HD$^{\Delta WHD}$, 2 and ssDNA (*Figure 4*).

## Achieving selectivity by Helical Hairpin interactions

In addition to the ARL, several other regions of BLM interact with 2 when it is bound to the ABS, including amino acids from helicase motifs I and III, plus a short region just upstream of motif IV (*Gorbalenya and Koonin, 1993*; *Hall and Matson, 1999*) (pre-motif IV, *Figure 4A*). Unsurprisingly, the amino acid sequence identity of each region across the RecQ-family is extremely high, and do not therefore provide a facile explanation for the observed selectivity of 2 (*Figure 7A*). In particular, two of the amino acid side chains involved in hydrogen bonds with 2 are absolutely conserved in identity (motif I, Ser801; motif III, Asp840, *Figures 7A* and *3C*). Likewise, amino acids within these motifs involved in hydrophobic contacts with 2 are also highly conserved in identity/chemical property. However, the third hydrogen bonding interaction (made by Asn1022 to 2) and its position within the 'Helical Hairpin' of the Zn-binding domain provides some insight, as both the length and amino acid composition of this loop is highly divergent across the RecQ-family and absent from RecQ4 (*Figure 7A*). There is no obvious consensus for any of the residues, in equivalent positions to those in BLM, involved in compound interaction. This observation provides a plausible route, although the addition or alteration of chemical groups to the core scaffold of 2, for generating potent and highly selective inhibitors for the individual members of the RecQ-family of helicases.

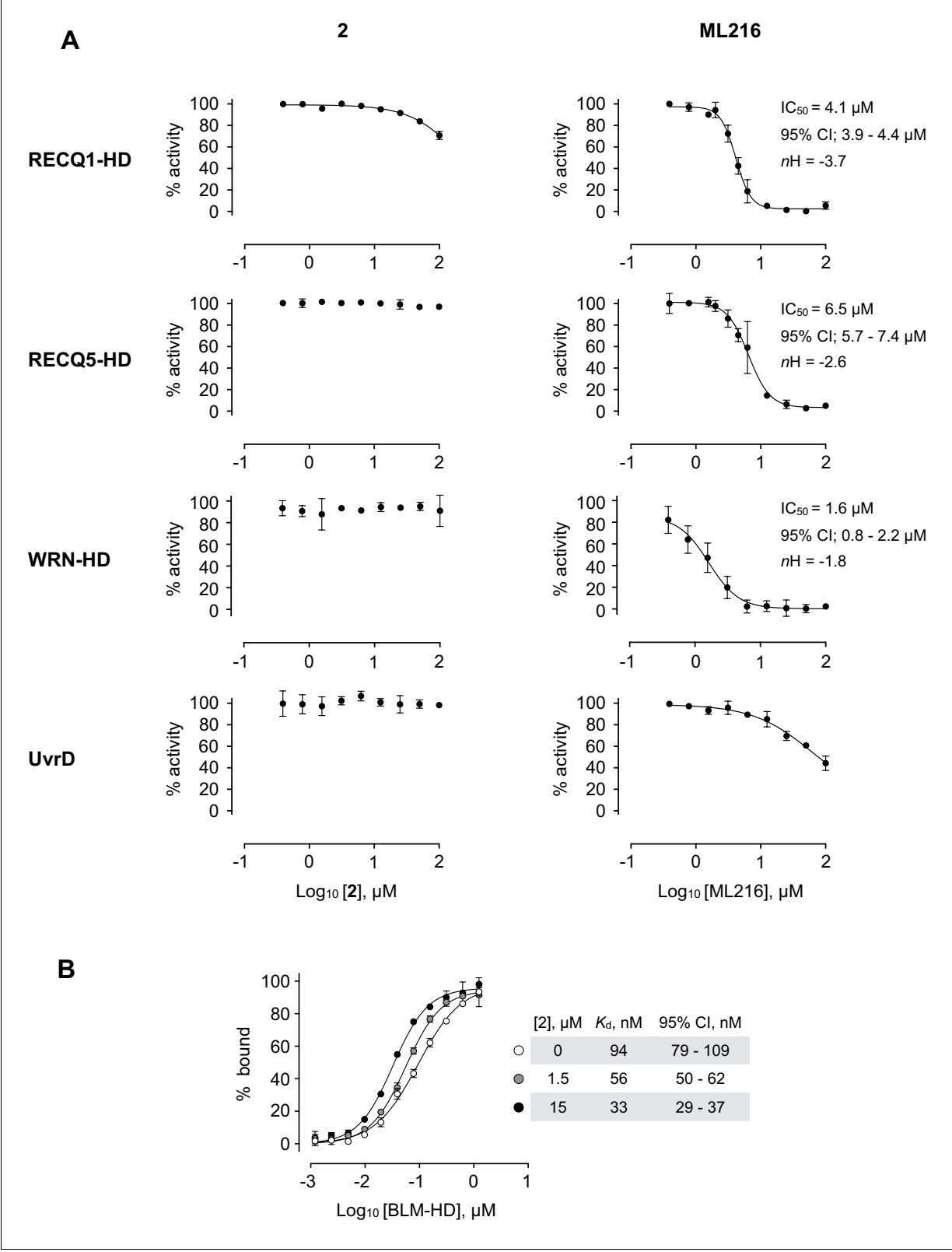

**Figure 6.** Selectivity profile of compound 2. (**A**) Dose response curves from ATP-turnover assays for titration of compounds 2 and ML216 against purified recombinant BLM-HD, WRN-HD, RecQ1-HD, RecQ5-HD and UvrD respectively. Calculated values for $IC_{50}$, $nH$ and 95% CI are given in each case. (**B**) MST-derived binding isotherms for the interaction of BLM-HD with ssDNA-20mer in the presence of increasing concentrations of 2. Calculated values for $K_d$ and 95% CI are given in each case. For all plots, data represent the mean of three technical replicates with error bars representing 1 SD.

*Figure 6 continued on next page*

*Figure 6 continued*

The online version of this article includes the following figure supplement(s) for figure 6:

**Figure supplement 1.** The active site of the liganded complex is compatible with binding of ATP, as judged by superposition of PDB entry 1OYY (*E. coli* RecQ in complex with ATP-γS *Bernstein et al., 2003*).

## Release of the HRDC

During the catalytic cycle, it is clear that the HRDC must be 'released' in order to interact with ssDNA, indeed amino acid residues within the aforementioned 'hydrophobic $3_{10}$ helix' contribute to both the 'parked' and 'engaged' interfaces; as exemplified by the side chain of Phe1238 that moves from a packing interaction with Pro956, a residue in the D1 domain, to an interaction with a base in the bound ssDNA-15mer (*Figure 5A*). As a consequence of the extended ssDNA interface visualised in our liganded complex (also seen in PDB entry 4TMU), we can see that the size and shape of the pocket that serves to bind the HRDC is substantially altered. Interaction of the D1 domain with ssDNA alters its spatial relationship with the D2 domain, leading to a widening of the pocket and to disruption of the hydrogen bonds that have previously been described to anchor the HRDC in place (*Newman et al., 2015*; *Figure 4—figure supplement 3*). Binding and release of the HRDC has previously been linked to nucleotide-status, indicating that the domain only becomes disengaged when BLM is not bound to ADP or ATP (*Newman et al., 2015*). Our data suggest an additional nuance: if the HRDC is able to engage in an ssDNA interaction, it is prevented from re-binding to the catalytic core, thus isolating it from the catalytic cycle of the enzyme.

## A speculative model for the involvement of the HRDC in unwinding DNA substrates

During model building and evaluation of our liganded structure, we found that the nucleic-acid interactions made by the different sub-domains of BLM-HD$^{\Delta WHD}$ explain how the HRDC might contribute to the ability of BLM to unwind different types of DNA substrate—incorporating information taken from our own structure, as well as that for the interaction of the WHD with dsDNA from PDB entry 4CGZ (*Figure 7B*).

Simple superposition of the two structures results in a clash of the HRDC (in our structure) with the WHD; however, this is readily resolved by a small horizontal translation of the HRDC (roughly equivalent to adding an additional nucleotide to the single-stranded portion of the bound DNA substrate). We also note that, in our liganded structure, the polarity of the single-stranded DNA interacting with the HRDC is opposite to that of our model but with the understanding that is actually dictated by the packing arrangement of the molecules that serve to generate the crystal lattice; on examination, each of the observed HRDC interactions is fully compatible with binding to ssDNA in either orientation.

However, using the trajectories for each of the bound DNA substrates as positional markers allows generation of a model for unwinding of a simple DNA duplex. The double-stranded portion of the substrate is held in place through the previously described set of interactions with the WHD domain (*Newman et al., 2015*). The β-hairpin of the WH serves to separate the DNA duplex, with one strand 'actively' engaged with the D1/D2 domains of the helicase core and the Helical Hairpin of the Zn-binding domain. The second 'inactive' strand passes along the opposite face of the WH β-hairpin, to subsequently interact with the HRDC domain, potentially acting to prevent reversion and re-annealing of the DNA duplex. The relatively poor binding affinity of the HRDC for ssDNA is compatible with this model, as it would allow iterative release and recapture of the 'inactive' strand as the DNA substrate is unwound.

A model for how BLM might unwind a Holliday Junction has previously been reported (*Kitano, 2014*) but this treats the HRDC domain as a static object, leaving in it the 'parked' position and making no interactions with nucleic acid. Data published subsequent to this paper has indicated that the HRDC plays a more fundamental role, for example the charge-reversal mutation N1239D to has been shown to ablate interaction of the HRDC with both ssDNA and a Holliday junction substrate (*Kim and Choi, 2010*). The HRDC has also been reported to confer DNA-structure specificity to BLM, with Lys1270 playing a role in mediating interactions with DNA and for efficient dissolution of double-Holliday junction substrates in vitro (*Wu et al., 2005*). Consistent with this, our structural

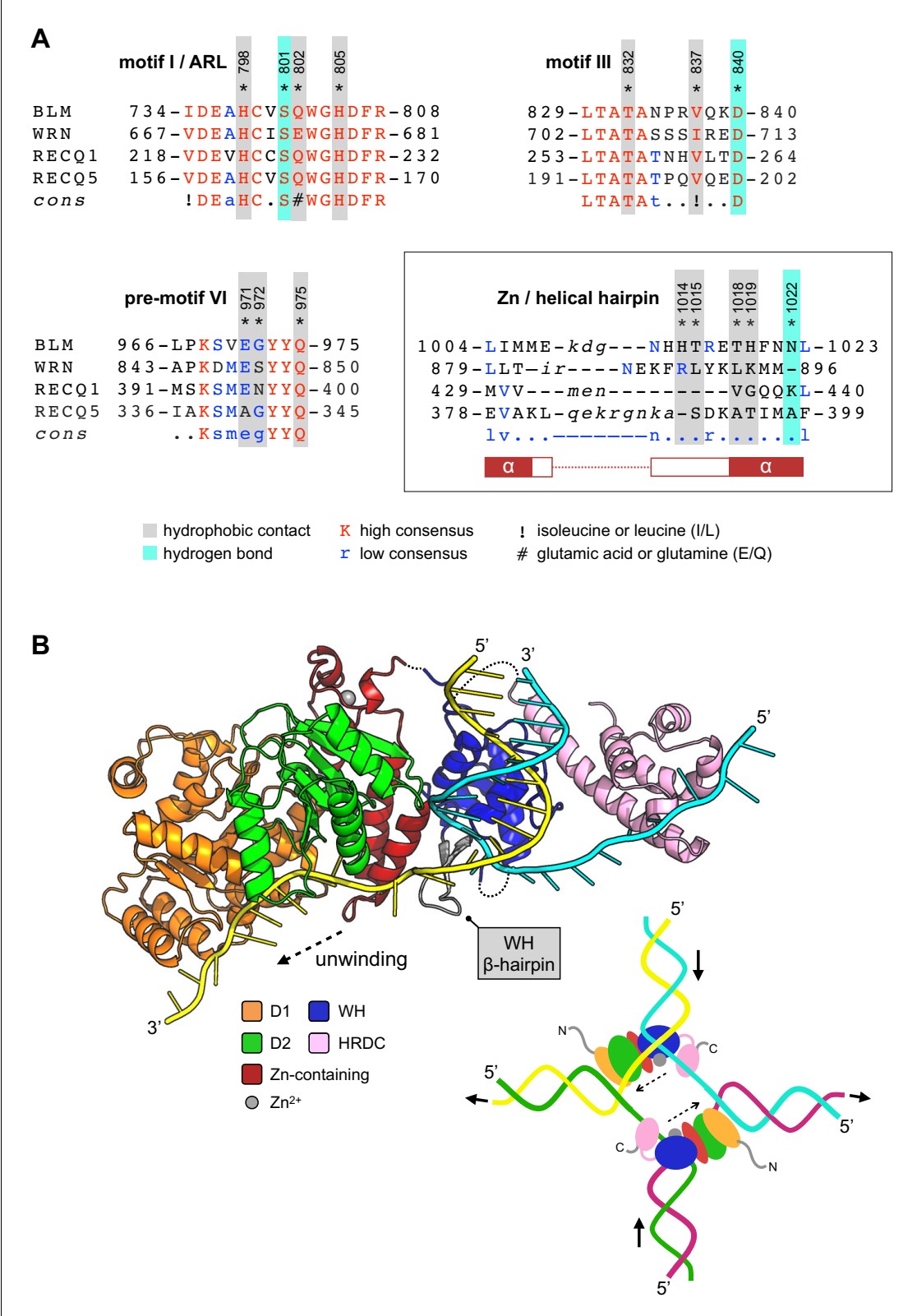

**Figure 7.** Compound selectivity/speculative model. (**A**) The observed selectivity of compound 2 appears to arise from interactions made with amino acids of the Zn-binding domain 'Helical Hairpin' that are poorly conserved (or absent) from the other RecQ-family helicases. For each multiple amino acid sequence alignment shown, highly conserved amino acids are coloured in red. Those conserved in at least two RecQ family members (low consensus) are coloured blue. Amino acids residues of human BLM involved in interactions with 2 are additionally highlighted. Please see associated

*Figure 7 continued on next page*

*Figure 7 continued*

key for additional information. (B) A speculative model for how the HRDC domain may contribute to the unwinding of a DNA duplex or a Holliday junction (inset) via transient interactions with the passive or 'inactive' strand. See associated key for additional details.

data reveals that both Asn1239 and Lys1270 of the HRDC are poised to interact with a section of single-stranded DNA just one to two nucleotides longer than that captured in our crystal lattice (*Figure 5B*). Finally, our model also suggests the existence of a transient direct interaction between the WHD and the HRDC.

Compound 2 itself, is unlikely to be suitable as a therapeutic agent, due to its poor ability to penetrate the cell membrane of mammalian cells (data not shown). Indeed, subsequent iterations of 2, or compounds that bind to the same allosteric site, may lead a tool compound that could be used to discover or confirm synthetic lethal relationships with BLM across different tumour backgrounds (*Datta et al., 2021*).

However, as the first described *bona fide* selective allosteric inhibitor of human BLM, the understanding of its mode of action will aid ongoing efforts to develop molecules targeting this class of enzymes for the treatment of human disease.

# Materials and methods

**Key resources table**

| Reagent type (species) or resource | Designation | Source or reference | Identifiers | Additional information |
|---|---|---|---|---|
| Gene (*Homo sapiens*) | BLM | UniProt | P54132 | BLM_HUMAN |
| Gene (*Homo sapiens*) | RECQL | UniProt | P46063 | RECQ1_HUMAN |
| Gene (*Homo sapiens*) | WRN | UniProt | Q14191 | WRN_HUMAN |
| Gene (*Homo sapiens*) | RECQL5 | UniProt | O94762 | RECQ5_HUMAN |
| Gene (*Escherichia coli*) | uvrD | UniProt | P03018 | UVRD_ECOLI |
| Strain (*Escherichia coli*) | BL21(DE3) | New England Biolabs | C2527I | Competent Cells |
| Sequence-based reagent | ssDNA-15mer | This paper | | CGTACCCGATGTGTT |
| Sequence-based reagent | ssDNA-20mer | This paper | | CGTACCCGATGTGTTCGTTC |
| Sequence-based reagent | Forked-50mer: FORK A | This paper | | XGAACGAACACATCGGGTACG TTTTTTTTTTTTTTTTTTTTTTTTTTTTTT X = Black Hole Quencher 2 or none |
| Sequence-based reagent | Forked-50mer: FORK B | This paper | | TTTTTTTTTTTTTTTTTTTTTTTTTTTTTT CGTACCCGATGTGTTCGTTCY Y = Tetramethylrhodamine or none |
| Recombinant DNA reagent | pET-17b | Novagen Merck Millipore | 69663 | |
| Recombinant DNA reagent | pNIC28-Bsa4 | Addgene | 26103 | |
| Chemical compound, drug | ML216 | Merck KGaA Caymen Chemical | SML0661 15186 | |
| Commercial assay or kit | PiColorLock Gold Phosphate Detection System | Novus Biologicals | 303–0030 | |
| Commercial assay or kit | DNA Unwinding Assay Kit | Inspiralis | DUKSR001 | |
| Software, algorithm | BUSTER | Global Phasing | RRID:SCR_015653 | |
| Software, algorithm | CCP4 | CCP4 | RRID:SCR_007255 | |

*Continued on next page*

*Continued*

| Reagent type (species) or resource | Designation | Source or reference | Identifiers | Additional information |
|---|---|---|---|---|
| Software, algorithm | Coot | Coot | RRID:SCR_014222 | |
| Software, algorithm | Fiji | Fiji | RRID:SCR_002285 | |
| Software, algorithm | Phaser | Phaser | RRID:SCR_014219 | |
| Software, algorithm | PHENIX | PHENIX | RRID:SCR_014224S | |
| Software, algorithm | Prism | GraphPad | RRID:SCR_002798 | |
| Software, algorithm | XDS | XDS | RRID:SCR_015652 | |

## Compound synthesis and purification

Details for synthesis and purification of compounds is provided in Appendix 1.

## Expression constructs

Synthetic genes, codon-optimised for expression in *E. coli*, were purchased from GeneArt (Thermo-Fisher Scientific, Loughborough, UK). With the exception of RECQ5 (see below) the coding sequence was subcloned into an in-house modified pET-17b vector at the NdeI and EcoRI sites of the multiple cloning site.

### BLM-HD, BLM-HD$^{\Delta WHD}$

pAWO-STREP-3C; a pET-17b expression vector modified to encode an N-terminal human rhinovirus 3C-protease (HRV-3C) cleavable StrepII-affinity tag. BLM-HD encodes amino acids 636–1298 of human BLM (UniProt ID: BLM_HUMAN), whereas BLM-HD$^{\Delta WHD}$ encodes amino acids 636–1074 and 1231–1298.

### RECQ1-HD

pAWO-His-TRX-3C; a pET-17b expression vector modified to encode an N-terminal His$_6$-tagged *E. coli* Thioredoxin HRV-3C cleavable affinity/solubility tag. RECQ1-HD encodes amino acids 49–616 of human RECQ1 (Uniprot ID: RECQ1_HUMAN).

### WRN-HD

pAWO-His-SUMO-3C; a pET-17b expression vector modified to encode an N-terminal His$_6$-tagged *S. cerevisiae* Smt3 (SUMO) HRV-3C cleavable affinity/solubility tag. WRN-HD encodes amino acids 480–1251 of human WRN (Uniprot ID: WRN_HUMAN).

### RECQ5-HD

The expression construct pNIC28-Bsa4-RecQL5 was obtained from the Structural Genomics Consortium, Oxford (see https://www.thesgc.org/tep/RECQL5 for full details). RECQ5-HD encodes amino acids 11–526 of human RECQ5 (Uniprot ID: RECQ5_HUMAN).

## Expression and purification

### BLM-HD, BLM-HD$^{\Delta WHD}$

*E. coli* strain BL21(DE3) [New England Biolabs, Hitchin, UK] was transformed with the required expression plasmid. A 'starter' culture was generated by inoculating a 250 ml glass Erlenmeyer flask with 100 ml of Turbo-broth [Molecular Dimensions, Sheffield, UK] supplemented with 50 μg/ml ampicillin. The culture was allowed to grow in an orbital-shaking incubator set at 37°C, 220 rpm, until the absorbance at 600 nm reached 1.5. The culture was then stored at 4°C until the following day. Twelve ml of the 'starter' culture was used to inoculate a 2 l Erlenmeyer containing 1lL of Turbo-broth supplemented with antibiotic as before. The culture was grown until the absorbance at 600 nm reached 1.5, when the flask containing the culture was placed on ice for a period of 30 min. During this time, the incubator temperature was reduced to 20°C. After incubation on ice, isopropyl-β-D-thiogalactoside (IPTG) was added to a final concentration of 0.4 mM, to induce protein

expression. The flask was then returned to the incubator, and the culture allowed to grow overnight at the reduced temperature of 20°C. Cells were harvested by centrifugation after a period of 16 hr. The resultant cell pellet was stored at −20°C until required.

The cell pellet arising from 2 l of culture was resuspended, on ice, in 25 ml of Buffer A (50 mM HEPES-NaOH, pH 7.5, 1 M NaCl, 0.5 mM TCEP, 0.5 mM EDTA) supplemented with a protease inhibitor tablet (cOmplete EDTA-free Protease Inhibitor Cocktail Tablet; Roche, Burgess Hill, UK). The cells were lysed by sonication and insoluble material removed by centrifugation. The resultant supernatant was applied to a 5 ml Strep-Tactin Superflow Plus Cartridge (Qiagen, Manchester, UK), pre-equilibrated with Buffer A. Unbound material was application of 10 column volumes (CV) of Buffer A (50 ml). Retained proteins were then eluted from the column by application of 5 CV of Buffer B (Buffer A supplemented with 5 mM desthiobiotin). Fractions containing the required protein were identified by SDS-PAGE, pooled, and then concentrated to a final volume of 2.5 ml using centrifugal concentrators (Vivaspin 20, 5000 MWCO; Sartorius Stedim Biotech GmBH, Goettingen, Germany). After overnight cleavage of the affinity tag with human rhinovirus 3C-protease, the sample was diluted to reduce the NaCl concentration to below 250 mM. This was applied to a 5 ml HiTrap Heparin HP cartridge (GE Healthcare Life Sciences, Little Chalfont, UK), pre-equilibrated in Buffer C (50 mM HEPES-NaOH, pH 7.5, 250 mM NaCl, 0.5 mM TCEP, 0.5 mM EDTA). Unbound material was removed by washing the column with 10 CV of buffer C. A linear NaCl gradient starting at a concentration of 250 mM and ending at 2000 mM, over 50 CV, was applied to the column. Fractions containing the desired recombinant protein were identified, pooled and concentrated as before. The concentrated sample was then applied to an HiLoad 26/600 Superdex 200 size exclusion chromatography column [GE Healthcare] pre-equilibrated in Buffer D (20 mM HEPES-NaOH pH7.5, 250 mM NaCl, 0.5 mM TCEP). Again, fractions containing the desired recombinant protein were identified, pooled and concentrated, then flash-frozen in aliquots in liquid nitrogen and stored at −80°C until required.

## RecQ1-HD, RecQ5-HD, WRN-HD

Expression and purification of RecQ1-HD, RecQ5-HD and WRN-HD were achieved using procedures similar to that used for BLM-HD, but with initial capture achieved using an IMAC column. Samples were applied to a HiTrap 5 ml TALON Crude column (GE Healthcare) pre-equilibrated in Buffer A (50 mM HEPES-NaOH pH 7.5, 500 mM NaCl, 0.5 mM TCEP, 10 mM imidazole). The column was washed with five column volumes of Buffer A, with retained protein eluted by the addition of 5 CV of Buffer B (50 mM HEPES-NaOH pH 7.5, 500 mM NaCl, 0.5 mM TCEP, 300 mM imidazole). Affinity/solubility tags were removed by incubation with either HRV-3C (RecQ1, WRN) or TEV protease (RecQ5).

## UvrD

Purified recombinant *E. coli* UvrD was kindly provided by Dr. Mohan Rajasekaran (Sussex Drug Discovery Centre, University of Sussex, UK).

## REAGENTS
### Solutions

Mg-ATP = 50 mM MgCl$_2$, 50 mM ATP
Mg-ADP = 50 mM MgCl$_2$, 50 mM ADP

### Oligonucleotides

Reverse-phase purified oligonucleotides were purchased from either Kaneka Eurogentec S.A. (Seraing, Belgium) or Eurofins Genomics Germany GmbH (Ebersberg, Germany).

*ssDNA-15mer:* 5'-CGTACCCGATGTGTT-3'
*ssDNA-20mer:* 5'-CGTACCCGATGTGTTCGTTC-3'
Forked-50mer
A: 5'-XGAACGAACACATCGGGTACGTTTTTTTTTTTTTTTTTTTTTTTTTTTTTTT-3'
B: 5'-TTTTTTTTTTTTTTTTTTTTTTTTTTTTTCGTACCCGATGTGTTCGTTCY-3'

Where X and Y are the following modifications:

Unwinding assay; X = BHQ2 (Black Hole Quencher 2), Y = TAMRA (tetramethylrhodamine)
Gel-based assay; X = none; Y = TAMRA
Dye-displacement assay; X = none; Y = none

FORK-A and FORK-B were annealed at a concentration of 200 µM using a slow-cooling cycle programmed into a PCR thermal cycler, in a buffer containing 20 mM HEPES.NaOH pH 7.5, 50 mM NaCl and 1 mM MgCl$_2$.

## Commercially sourced ML216

ML216 was purchased from Merck KGaA (Darmstadt, Germany), product code: SML0661. ML216-A was purchased from Cayman Chemical (Ann Arbor, Michigan, USA), product code: 15186.

## Biochemical assays

### Fluorescence-based DNA unwinding assay

Methodology is based on that previously reported by *Rosenthal, 2010*. Briefly, assays were carried out in 384-well black plates, with measurements taken at emission and excitation wavelengths of 540 and 590 nm respectively, in a PHERAstar multimode plate reader (BMG Labtech). Assay buffer: 50 mM Tris-HCl pH 8.0, 50 mM NaCl, 2 mM MgCl$_2$, 0.01% v/v Tween-20, 2.5 µg/ml poly(dI-dC), 1 mM DTT.

A total of 28 µl of BLM-HD (at 0.535 nM in assay buffer) was pre-incubated with 2 µl of compound (2 mM stock dissolved in 100% v/v DMSO, over a range of final concentrations up to 100 µM) for a period for 15 min at room temperature. Next, 10 µl of substrate (40 nM forked-50mer dsDNA and 2000 µM Mg-ATP) was added, then incubated for a further 20 min at room temperature, before the final fluorescent intensity for each well was measured.

Assay conditions (compounds 2 to 7 and ML216): 0.375 nM BLM-HD, 10 nM annealed DNA substrate, 500 µM Mg-ATP in a reaction volume of 40 µl over a 20 min incubation period.

Data for compound 1 are taken from an earlier iteration of the assay and were measured used the conditions: 3.75 nM BLM-HD, 75 nM annealed DNA substrate, 120 µM Mg-ATP in a reaction volume of 40 µl over a 20-min incubation period.

## Malachite-green ATP turnover assay

Assay uses the PiColorLock Gold Phosphate Detection System from Novus Biologicals following the manufacturer's recommended protocol. Briefly, assays were carried out in 96-well clear flat-bottomed plates, with absorbance measurements taken at a wavelength of 630 nm in a CLARIOstar multimode plate reader (BMG Labtech). Assay buffer: 50 mM Tris-HCl pH 7.5, 50 mM NaCl, 2 mM MgCl$_2$, 0.05% v/v Tween-20, 0.5 mM TCEP.

165 µl of BLM-HD and ssDNA-20mer (at a concentration of 2.4 nM and 121 nM, respectively) was pre-incubated with 10 µl of compound (2 mM stock dissolved in 100 % v/v DMSO, over a range of final concentrations up to 100 µM) for a period of 15 min at room temperature. Next, 25 µl of Mg-ATP substrate (at 16 mM) was added. After 20 min, reactions were stopped by the addition of 50 µl Gold mix (a 100:1 ratio of PiColorLock:Accelerator reagents). After 2 min, 20 µl of stabiliser solution was added to each well. After a further 30 min absorbance measurements were taken.

Assay conditions: 2 nM BLM-HD, 100 nM ssDNA-20mer and 2 mM Mg-ATP in a reaction volume of 200 µl over a 20-min incubation period.

## Gel-based assay

Assay buffer: 50 mM Tris-HCl pH 8.0, 50 mM NaCl, 2 mM MgCl$_2$, 0.01% v/v Tween-20, 2.5 µg/ml poly(dI-dC), 1 mM DTT. 28 µl of BLM-HD (at a concentration of 2.9 nM) was pre-incubated with 2 µl of compound (2 mM stock dissolved in 100 % v/v DMSO, over a range of concentrations up to 100 µM) for a period of 15 min at room temperature. Next, 10 µl of substrate (300 nM forked-50mer dsDNA and 4.8 mM Mg-ATP) were added. After 10 min, reactions were terminated by the addition of 1 x loading dye (6 x solution: 10 mM Tris-HCl pH 7.5, 0.03% w/v bromophenol blue, 60% v/v glycerol, 60 mM EDTA). The samples were then loaded onto a 15% native gel (29:1 acrylamide:bis-acrylamide, 0.5 x TBE), separated by electrophoresis, and then visualised using a FLA-1500 fluorimager

[Fujifilm, Bedford, UK]. The intensity of each species on the gel was quantified using the analysis tools provided in the software package Fiji (*Schindelin et al., 2012*).

## Topoisomerase I DNA-unwinding assay

Assay uses the DNA Unwinding Assay Kit from Inspiralis (Norwich, UK) following the manufacturer's recommended protocol. Resultant samples were applied to a 1% w/v agarose gel (in 1 x TAE buffer), separated by electrophoresis, stained with ethidium bromide, and then visualised with a UV-transilluminator/digital gel documentation system.

## Dye-displacement assay

Fluorescence intensity was measured in a CLARIOstar multi-mode plate reader (BMG Labtech) with excitation and emission wavelengths of 485 nm and 520 nm respectively, in 384-well black plates. Twenty-eight µl of forked-50mer dsDNA (at a final concentration of 800 nM) was pre-incubated with 10 µl of SYBR Green II (1:200 dilution) for a period of 20 min at room temperature. Two µl of compound (1 mM stock dissolved in 100 % v/v DMSO, over a range of final concentrations up to 50 µM) was then added. Measurements were taken after incubation times of 20, 45, and 60 min. Assay buffer: 50 mM Tris-HCl pH 8.0, 50 mM NaCl, 2 mM $MgCl_2$, 0.01% v/v Tween-20, 1 mM DTT.

Assay conditions: 800 nM annealed DNA substrate and 1:800 SYBR Green II in a reaction volume of 40 µl over a 20-min incubation period.

## Biophysical assays

### Microscale thermophoresis (MST)

Experiments were performed in a Monolith NT.115 instrument from NanoTemper Technologies GmbH (München, Germany). Purified recombinant protein was labelled using a Monolith NT RED-Maleimide Protein Labelling Kit supplied by the manufacturer, following the recommended protocol. A total of 19 µl of BLM-HD (at a final concentration of 75 nM) was mixed with 1 µl of the required 'ligand' solution (ssDNA and / or compound) and incubated for 15 min at room temperature, before being transferred to 'premium' capillaries for measurement. Experiments were performed at a temperature of 25°C, with settings of 20% excitation power, 20% MST power. Assay buffer: 50 mM Tris-HCl pH 7.5, 100 mM NaCl, 2 mM $MgCl_2$, 0.05% v/v Tween-20, 0.5 mM TCEP.

## Crystallography

### BLM-HD$^{\Delta WHD}$ / ADP

Prior to setting up of crystallisation screens BLM-HD$^{\Delta WHD}$ at a concentration of 15 mg/ml was combined with glycerol (100% v/v) and Mg-ADP (50 mM) to produce final concentrations of 10% v/v and 2 mM, respectively. Of the prepared complex, 150 nl was combined with 150 nl of crystallisation reagent in MRC2 sitting drop vapour diffusion experiments against a reservoir volume of 50 µl. Crystals were obtained in condition A8 of the Morpheus HT-96 screen (0.06 M divalents, 37.5% Buffer System 2% and 37.5% Precipitant Mix 4); Molecular Dimensions [Sheffield, UK] at 4°C after a period of approximately 1 week.

> Divalents = 0.3M magnesium chloride, 0.3M calcium chloride
> Buffer system 2 = 1M sodium HEPES, MOPS (acid) pH 7.5
> 75% Precipitant Mix 4 = 25% w/v MPD, 25% v/v PEG1000, 25% w/v PEG 3350

Cryoprotection for data collection was achieved by stepwise soaking of crystals in buffers containing increasing amounts of ethylene glycol, to a final concentration of 20% (v/v). Diffraction data to a resolution of 1.53 Angstrom were collected from a single crystal, on beamline I04 at the Diamond Light Source (Didcot, UK). Crystals were in space group P2$_1$ with one molecule of BLM-HD$^{\Delta WHD}$ plus associated ligands forming the asymmetric unit.

### BLM-HD$^{\Delta WHD}$ / ADP / ssDNA-15mer / compound 2

BLM-HD$^{\Delta WHD}$ was mixed with ssDNA-15mer at a 1:1.2 molar ratio (protein:DNA) to produce a final concentration of 15 mg/ml with respect to protein. Compound **2** was then added to a final concentration of 3 mM (from a stock at 100 mM in 100% v/v DMSO) and incubated with the protein:DNA complex overnight at 4°C. Prior to setting up crystallisation trials the complex was combined with

glycerol (100% v/v) and Mg-ADP (50 mM) to produce final concentrations of 10% v/v and 2 mM, respectively. 150 nl of the prepared complex was combined with 150 nl of crystallisation reagent in MRC2 sitting drop vapour diffusion experiments against a reservoir volume of 50 µl. Crystals were obtained in condition C9 of the Morpheus HT-96 screen (0.09 M NPS, 0.1M Buffer System, 30% Precipitant Mix 1, Molecular Dimensions) at 4°C after a period of approximately 1 week.

> NPS = 0.3 M sodium nitrate, 0.3 M sodium phosphate dibasic, 0.3 M ammonium sulphate
> Buffer System 1 = 1.0 M imidazole, MES monohydrate (acid) pH 6.5
> 60% Precipitant Mix 1 = 40% v/v PEG 500 MME, 20% w/v PEG 20000

Cryoprotection for data collection was achieved by stepwise soaking of crystals in buffers containing increasing amounts of ethylene glycol, to a final concentration of 20% (v/v). Diffraction data to a resolution of 3.0 Angstrom were collected from a single crystal, on beamline I03 at the Diamond Light Source (Didcot, UK). Crystals were in space group P1 with six molecules of BLM-HD$^{\Delta \text{WHD}}$ plus associated ligands forming the asymmetric unit.

### Data processing and model building

Diffraction data were automatically processed at the synchrotron by the xia2 pipeline (*Winter et al., 2013*), using software packages DIALS (*Beilsten-Edmands et al., 2020*; *Winter et al., 2018*) or XDS (*Kabsch, 2010*) and Aimless (*Winn et al., 2011*). For BLM-HD$^{\Delta \text{WHD}}$/ADP, coordinates corresponding to the helicase domain were extracted from PDB entry 4O3M and provided as a search model for molecular replacement using Phaser (*McCoy et al., 2007*). For BLM-HD$^{\Delta \text{WHD}}$/ADP/ssDNA-15mer/ compound 2, the rebuilt and refined coordinates for BLM-HD$^{\Delta \text{WHD}}$ were used as the search model. Initial models were extended and improved by iterative rounds of building in Coot (*Emsley and Cowtan, 2004*) and refinement in either PHENIX (*Liebschner et al., 2019*) or BUSTER (*Bricogne, 2020*) to produce the final deposited models. Crystallisation and refinement statistics are provided in *Appendix-table 1*.

### Data plotting and analysis

All experimental data were plotted and analysed using *GraphPad Prism, 2020*.

## Acknowledgements

General: We thank Will Pearce (Sussex Drug Discovery Centre) for discussions relating to assay development. We are also grateful to the Diamond Light Source Ltd., Didcot, UK for access to synchrotron radiation. Funding: This work was supported by funding from: Chinese Scholarship Council (XC and FGMP), Wellcome Trust 110578/Z/15/Z (SEW), Wellcome Trust 105630/Z/14/Z (FGMP, AWO and LHP), Cancer Research UK Programme Grant C302/A24386 (LHP and AWO).

## Additional information

### Funding

| Funder | Grant reference number | Author |
| --- | --- | --- |
| Cancer Research UK | C302/A24386 | Laurence H Pearl<br>Antony W Oliver |
| China Scholarship Council | | Xiangrong Chen<br>Frances MG Pearl |
| Wellcome | 105630/Z/14/Z | Laurence H Pearl<br>Frances MG Pearl<br>Antony W Oliver |
| Wellcome | 110578/Z/15/Z | Simon E Ward |

The funders had no role in study design, data collection and interpretation, or the decision to submit the work for publication.

## Author contributions
Xiangrong Chen, Formal analysis, Funding acquisition, Investigation, Methodology, Writing - review and editing; Yusuf I Ali, Jessica R Booth, Jessica JR Hudson, Investigation, Methodology; Charlotte EL Fisher, Investigation; Raquel Arribas-Bosacoma, Supervision, Investigation; Mohan B Rajasekaran, S Mark Roe, Supervision, Investigation, Methodology; Gareth Williams, Sarah Walker, Supervision, Methodology; Laurence H Pearl, Funding acquisition, Investigation, Methodology, Writing - original draft, Writing - review and editing; Simon E Ward, Conceptualization, Supervision, Funding acquisition, Investigation, Methodology, Writing - review and editing; Frances MG Pearl, Conceptualization, Supervision, Funding acquisition, Investigation; Antony W Oliver, Conceptualization, Data curation, Formal analysis, Supervision, Funding acquisition, Validation, Investigation, Visualization, Methodology, Writing - original draft, Writing - review and editing

## Author ORCIDs
Xiangrong Chen ⓘ https://orcid.org/0000-0003-0587-3444
Raquel Arribas-Bosacoma ⓘ https://orcid.org/0000-0002-2884-1253
Jessica JR Hudson ⓘ https://orcid.org/0000-0003-3189-2622
S Mark Roe ⓘ https://orcid.org/0000-0002-7371-9855
Laurence H Pearl ⓘ http://orcid.org/0000-0002-6910-1809
Simon E Ward ⓘ https://orcid.org/0000-0002-8745-8377
Frances MG Pearl ⓘ https://orcid.org/0000-0002-8210-4393
Antony W Oliver ⓘ https://orcid.org/0000-0002-2912-8273

## Decision letter and Author response
Decision letter https://doi.org/10.7554/eLife.65339.sa1
Author response https://doi.org/10.7554/eLife.65339.sa2

# Additional files
## Supplementary files
• Transparent reporting form

## Data availability
Diffraction data have been deposited in the PDB under the accession codes 7AUC and 7AUD.

The following datasets were generated:

| Author(s) | Year | Dataset title | Dataset URL | Database and Identifier |
|---|---|---|---|---|
| Chen X, Ali Y, Fisher CEL, Arribas-Bosacoma R, Rajasekeran MB, Williams G, Walker S, Roe SM, Pearl LH, Ward SE, Pearl FGM, Oliver AW | 2020 | Crystal structure of an engineered helicase domain construct for human Bloom syndrome protein (BLM) | https://www.rcsb.org/structure/7AUC | RCSB Protein Data Bank, 7AUC |
| Chen X, Ali Y, Fisher CEL, Arribas-Bosacoma R, Rajasekeran MB, Williams G, Walker S, Roe SM, Pearl LH, Ward SE, Pearl FGM, Oliver AW | 2020 | Structure of an engineered helicase domain construct for human Bloom syndrome protein (BLM) | https://www.rcsb.org/structure/7AUD | RCSB Protein Data Bank, 7AUD |

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

## Appendix 1

### Synthetic preparation of test molecules

All reactions were conducted under an atmosphere of nitrogen unless otherwise stated. Anhydrous solvents were used as purchased or were purified under nitrogen as follows using activated molecular sieves. Thin layer chromatography was performed on glass plates pre-coated with Merck silica gel 60 F254. Visualisation was achieved with U.V. fluorescence (254 nm) or by staining with a phosphomolybdic acid dip or a potassium permanganate dip. Flash column chromatography was carried out using pre-packed columns filled with Aldrich silica gel (40–63 μm) on an ISCO Combiflash Rf, or a Biotage Isolera Prime. Proton nuclear magnetic resonance spectra were recorded at 500 MHz on a Varian 500 spectrometer (at 30°C), using residual isotopic solvent ($CHCl_3$, $\delta H = 7.27$ ppm, DMSO $\delta H = 2.50$ ppm, 3.33 ppm ($H_2O$)) as an internal reference. Chemical shifts are quoted in parts per million (ppm). Coupling constants (J) are recorded in Hertz (Hz). Carbon nuclear magnetic resonance spectra were recorded at 125 MHz on a Varian 500 spectrometer and are proton decoupled, using residual isotopic solvent ($CHCl_3$, $\delta C = 77.00$ ppm, DMSO $\delta C = 39.52$ ppm) as an internal reference. Carbon spectra assignments are supported by HSQC and DEPT editing and chemical shifts ($\delta C$) are quoted in ppm. Infrared spectra were recorded on a Perkin Elmer FT-IR One spectrometer as either an evaporated film or liquid film on sodium chloride plates. Absorption maxima are reported in wave numbers ($cm^{-1}$). Only significant absorptions are presented in the data, with key stretches identified in brackets. LCMS data was recorded on a Waters 2695 HPLC using a Waters 2487 UV detector and a Thermo LCQ ESI-MS. Samples were eluted through a Phenomenex Lunar 3μ C18 50 mm ×4.6 mm column, using acetonitrile and water acidified by 0.01% formic acid in three methods: method 1 (3:7 to 7:3 acetonitrile and water over 7 min), method 2 (3:7 to 7:3 acetonitrile and water over 4 min) and method 3 (19:1 to 1:19 acetonitrile and water over 10 min), High resolution mass spectrometry (HRMS) spectra were recorded on Bruker Daltonics Apex III ESI-MS, with an Apollo ESI probe using a methanol spray. Only molecular ions, fractions from molecular ions and other major peaks are reported as mass/charge (m/z) ratios.

### *N*-(2-methyl-5-sulfamoyl-phenyl)−4-(2-methylthiazol-4-yl)benzamide (1)

#### Methyl 4-(2-bromoacetyl)benzoate

A solution of methyl 4-acetylbenzoate (1.00 g, 5.61 mmol) and *p*-toluenesulfonic acid monohydrate (54 mg, 0.28 mmol) in acetonitrile (30 mL) was treated with *N*-bromosuccinimide (0.99 g, 5.61 mmol) and the reaction mixture heated to 80°C for 16 hr. The solvent was removed under reduced pressure. The crude product was taken up in saturated aq. $NaHCO_3$ (15 mL) and extracted with ethyl acetate (3 × 15 mL). The combined organic components were then washed with brine (15 mL), dried over $MgSO_4$, filtered and concentrated under reduced pressure. The crude product was purified by column chromatography (silica 24 g, 0% to 10% ethyl acetate in petroleum ether) to yield the desired compound as a light yellow solid (1.12 g, 70%). Rf 0.66 (petroleum ether:ethyl acetate, 9:1); $^1$H-NMR (500 MHz, DMSO-d6) δ 8.14–8.01 (m, 4H, H-2, H-3), 4.98 (s, 2H, $COCH_2Br$), 3.89 (s, 3H, $COOCH_3$).

#### Methyl 4-(2-methylthiazol-4-yl)benzoate

To methyl 4-(2-bromoacetyl)benzoate (2.12 g, 8.23 mmol) in *N,N*-dimethylformamide (30 mL) was added thioacetamide (931 mg, 12.4 mmol) and the reaction mixture stirred at ambient temperature for 16 hr. Upon completion, water was added to the reaction mixture. The resulting precipitate was collected by vacuum filtration and dried under reduced pressure to afford the desired compound as a white solid (1.48 g, 73%). m.p. 223–225°C; $^1$H-NMR (500 MHz, DMSO-d6) δ 8.14 (s, 1H, H-6), 8.08 (d, J 8.2, 2H, H-2), 8.01 (d, J 8.2, 2H, H-3), 3.87 (s, $OCH_3$), 2.73 (s, $CH_3$); $^{13}$C-NMR (126 MHz, DMSO-d6) δ 166.5 (CO), 166.4 (C-8), 153.0 (C-5), 138.9 (C-4), 130.2 (C-2), 129.1 (C-1), 126.5 (C-3), 116.8 (C-6), 52.5 ($OCH_3$), 19.4 ($CH_3$); IR (neat, $\nu_{max}$, $cm^{-1}$) 3106, 2943, 1713, 1606, 1436, 1409, 1270, 1170; LCMS (LCQ) Rt = 2.9 min (method 2), m/z (ESI+) 234.2 $[M+H]^+$; HRMS m/z (ESI): calcd. for $C_{12}H_{11}NO_2S$ $[M+H]^+$ 234.0583, found 234.0583.

## 4-(2-Methylthiazol-4-yl)benzoic acid

To methyl 4-(2-methylthiazol-4-yl)benzoate (1.46 g, 6.26 mmol) in methanol (24 mL) and water (8 mL) was added sodium hydroxide (502 mg, 12.5 mmol) and the reaction mixture stirred at an ambient temperature for 16 hr. Upon completion, the reaction mixture was acidified to pH of 2–3 using 2 M aq. hydrochloric acid solution. The resulting precipitate was collected by vacuum filtration and dried under reduced pressure to afford the desired acid as a white solid (1.21 g, 84%). m.p. 250–252°C; $^1$H-NMR (500 MHz, DMSO-d6) δ 8.10 (s, 1H, H-6), 8.05 (d, J 8.2, 2H, H-3), 7.99 (d, J 8.1, 2H, H-2), 2.73 (s, 3H, CH3); $^{13}$C-NMR (126 MHz, DMSO-d6) δ 167.5 (CO), 166.4 (C-8), 153.2 (C-5), 138.5 (c-4), 130.3 (C-2), 126.4 (C-3), 116.5 (C-6), 19.4 (12-CH3). Quaternary 5 C not visible; IR (neat, $\nu_{max}$, cm$^{-1}$) 2826, 1667, 1608, 1573, 1421, 1290, 1169; 256 LCMS (LCQ) Rt = 0.6 min (method 1), m/z (ESI+) 220.2 [M+H]$^+$; HRMS m/z (ESI): calcd. for $C_{11}H_9NO_2S$ [M+H]$^+$ 220.0427, found 220.0428.

## N-(2-methyl-5-sulfamoyl-phenyl)-4-(2-methylthiazol-4-yl)benzamide (1)

To 4-(2-methylthiazol-4-yl)benzoic acid (120 mg, 0.55 mmol), HBTU [(2-(1H-benzotriazol-1-yl)−1,1,3,3-tetramethyluronium hexafluorophosphate] (153 mg, 0.66 mmol), N,N-diisopropylethylamine (191 μL, 1.09 mmol)) in N,N-dimethylformamide (2 mL) was added 3-amino-4-methylbenzenesulfonamide (102 mg, 0.55 mmol). The reaction mixture was stirred at ambient temperature for 16 hr. Upon completion, the solvent was removed under reduced pressure. The crude product was taken up in ethyl acetate (5 mL), washed with saturated aq. NaHCO$_3$ (4 mL), brine (4 mL), dried over MgSO$_4$, filtered and concentrated under reduced pressure. The crude was purified by column chromatography (silica 12 g, 0% to 55% ethyl acetate in petroleum ether) and further purified by column chromatography (amino silica 4 g, 0% to 5% methanol in dichloromethane) to yield the desired amide 1 as a white solid (15 mg, 7%). Rf 0.14 (petroleum ether:ethyl acetate 9:11); m.p. 261–263°C; $^1$H-NMR (500 MHz, DMSO-d6) δ 10.11 (s, 1H, CONH), 8.15–8.09 (m, 3H, H-3, H-6), 8.06 (d, J 8.2, 2H, H-2), 7.87 (s, 1H, H-6'), 7.62 (d, J 8.0, 1H, H-4'), 7.48 (d, J 8.0, 1H, H9'), 7.34 (s, 2H, SO$_2$NH$_2$), 2.75 (s, 3H, 8-CH$_3$), 2.32 (s, 3H, 2'-CH$_3$); $^{13}$C-NMR (126 MHz, DMSO-d6) δ 166.4 (CO), 165.5 (C-8), 153.3 (C-5), 142.5 (C-5'), 138.3 (C-1), 137.6 (C-2'), 137.2 (C-2'), 133.6 (C-4), 131.2 (C-3'), 128.75 (C-2), 126.3 (C3), 124.1 (C-6'), 123.4 (C-4'), 116.1 (C-6), 19.4 (8-CH3), 18.44 (2'-CH3); IR (neat, $\nu_{max}$, cm$^{-1}$) 3258, 2923, 1630, 1572, 255 1516, 1444, 1403, 1304, 1154; LCMS (LCQ) Rt = 2.7 min (method 1), m/z (ESI+) 388.1 [M+H]$^+$; HRMS m/z (ESI): calcd. for $C_{18}H_{17}N_3O_3S_2$ [M+H]$^+$ 388.6847, found 388.6850.

## *N*-(2,3-dimethyl-5-sulfamoyl-phenyl)-4-(2-methylthiazol-4-yl)benzamide (2)

To 4-(2-methylthiazol-4-yl)benzoic acid (120 mg, 0.55 mmol), HBTU (153 mg, 0.66 mmol), N,N-diisopropylethylamine (0.19 mL, 1.09 mmol) in N,N-dimethylformamide (3 mL) was added, 3-amino-4,5-dimethylbenzenesulfonamide (110 mg, 0.55 mmol). The reaction mixture was stirred at ambient temperature for 16 hr. Upon completion, the solvent was removed under reduced pressure. The crude product was taken up in ethyl acetate (5 mL), washed with saturated aq. NaHCO$_3$ (4 mL), brine (4 mL), dried over MgSO$_4$, filtered and concentrated under reduced pressure. The crude was purified by column chromatography (silica 12 g, 0% to 60% ethyl acetate in petroleum ether) to yield the desired amide 2 as a white solid (15 mg, 7%). Rf 0.12 (petroleum ether:ethyl acetate 1:1); m.p. 246–248°C; $^1$H-NMR (500 MHz, DMSO-d6) δ 10.17 (s, 1H, CONH), 8.13–8.08 (m, 3H, H-3, H-6), 8.05 (d, J 8.4, 2H, H-2), 7.64 (d, J 1.9, 1H, H-4'), 7.54 (d, J 1.9, 1H, H-3'), 7.28 (s, 2H, SO$_2$NH$_2$), 2.74 (s, 3H, 8-CH$_3$), 2.36 (s, 3H, 3'-CH$_3$), 2.17 (s, 3H, 2'-CH$_3$); $^{13}$C-NMR (126 MHz, DMSO-d6) δ 166.4 (CO), 165.7 (C-8), 153.3 (C-5), 141.6 (C-5'), 138.5 (C-1), 137.6 (ArC), 137.5 (ArC), 137.1 (ArC), 133.7 (C-4), 128.7 (C-2), 126.3 (C-3), 124.6 (C-6'), 122.3 (C-4'), 116.1 (C-6), 20.7 (3'-CH3), 19.4 (8-CH3), 15.0 (2'-CH3); IR (neat, $\nu_{max}$, cm$^{-1}$) 3258, 2923, 1630, 1572, 1516, 1444, 1403, 1304, 1154; LCMS (LCQ) Rt = 2.0 min (method 1), m/z (ESI+) 402.1 [M+H]$^+$; HRMS m/z (ESI): calcd. for $C_{19}H_{19}N_3O_3S_2$ [M+H]$^+$ 401.0868, found 401.0866

## *N*-(3-hydroxy-2-methyl-phenyl)−4-(2-methylthiazol-4-yl)benzamide (3)

To 4-(2-methylthiazol-4-yl)benzoic acid (80 mg, 0.36 mmol) in dichloromethane (3 mL) was added oxalyl chloride (37 μL, 0.44 mmol) in a dropwise manner followed by the addition of a few drops of

N,N-dimethylformamide (10 μL). The reaction mixture was stirred at ambient temperature for 16 hr. The crude acyl chloride was then added to a stirred mixture of 3-amino-2-methylphenol (54 mg, 0.44 mmol), N,N-diisopropylethylamine (397 μL, 2.28 mmol) and dichloromethane (1 mL) and stirred at ambient temperature for 2–16 hr. The solvent was removed under reduced pressure. The crude product was taken up in saturated aq. $NaHCO_3$ (5 mL) and extracted with ethyl acetate (3 × 10 mL). The combined organic components were then washed with brine (10 mL), dried over $MgSO_4$, filtered and concentrated under reduced pressure. The crude was purified by column chromatography (silica 12 g, 0% to 40% ethyl acetate in petroleum ether) to yield the desired amide three as a white solid (13 mg, 10%).Rf 0.47 (petroleum ether:ethyl acetate 1:1); m.p. 238–240°C; [1]H-NMR (500 MHz, DMSO-d6) δ 9.84 (s, 1H, CONH), 9.35 (s, 1H, OH), 8.10 (s, 1H, H-6), 8.07 (d, J 8.2, 2H, H-3), 8.03 (d, J 8.3, 2H, H-2), 7.00 (t, J 7.9, 1H, H-5'), 6.79 (d, J 7.8, 1H, H-6'), 6.73 (d, J 8.0, 1H, H-4'), 2.74 (s, 3H, 8-CH3), 2.03 (s, 3H, 2'-CH3); [13]C-NMR (126 MHz, DMSO-d6) δ 166.4 (CONH), 165.3 (C-8), 156.2 (C-5), 153.4 (C-3'), 137.9 (ArC), 137.3 (ArC), 134.1 (ArC), 128.7 (H-2), 126.2 (H-3), 126.0 (ArC), 121.3 (ArC), 118.0 (C-6'), 116.0 (C-6), 112.9 (C-4'), 19.4, (C-8) 11.4 (C-2'); IR (neat, $v_{max}$, cm$^{-1}$) 3291, 1640, 1607, 1499, 1466, 1307, 1174; LCMS (LCQ) Rt = 2.5 min (method 1), m/z (ESI+) 325.1 [M+H]$^+$; HRMS (ESI): calcd. for $C_{18}H_{16}NaN_2OS$ [M+Na]$^+$ 347.0825, found 347.0827.

## Methyl 4-methyl-3-((4-(2-methylthiazol-4-yl)benzoyl)amino)benzoate (4)

To 4-(2-methylthiazol-4-yl)benzoic acid (800 mg, 3.65 mmol) and methyl 3-amino-4-methylbenzoate (52 μL, 9.12 mmol) in tetrahydrofuran (15 mL) was added phosphorus trichloride (0.32 mL, 3.65 mmol). The reaction mixture was heated in a microwave reactor for 20 min at 150°C. The crude product was taken up in saturated aq. $NaHCO_3$ (5 mL) and extracted with ethyl acetate (3 × 10 mL). The combined organic components were then washed with brine (10 mL), dried over $MgSO_4$, filtered and concentrated under reduced pressure. The crude was purified by column chromatography (amino silica 12 g, 0% to 50% ethyl acetate in petroleum ether) to yield the desired amide 4 as a white solid (920 mg, 65%). Rf 0.23 (petroleum ether:ethyl acetate 3:1); m.p. 179–181°C; [1]H-NMR (500 MHz, DMSO-d6) δ 10.03 (s, 1H, CONH), 8.14–8.07 (m, 3H, H-3, H-6), 8.05 (d, J 8.3, 2H, H-2), 8.00 (d, J 1.8, 1H, H-6'), 7.76 (dd, J 7.9, 1.8, 1H, H-4'), 7.44 (d, J 7.9, 1H, H-9'), 3.85 (s, 3H, OCH3), 2.74 (s, 3H, 8-CH3), 2.33 (s, 3H, 2'-CH3); [13]C-NMR (126 MHz, DMSO-d6) δ 166.4 (CO), 165.6 (C-8), 153.3 (C-5), 140.0 (C-8'), 137.6 (C-1), 137.3 (C-7'), 133.7 (C-4), 131.3 (C-3'), 128.8 (C-2), 128.1 (C-6'), 127.5 (C-5'), 126.9 (C-4'), 126.3 (C-3), 116.1 (C-6), 52.5 (OCH3), 19.4 (8-CH3), 18.6 (2'-CH3). COO not visible; IR (neat, $v_{max}$, cm$^{-1}$) 3256, 1726, 1643, 1523, 1432, 1296, 1171, 1110; LCMS (LCQ) Rt = 2.6 min (method 1), m/z (ESI+) 367.0 [M+H]$^+$; HRMS m/z (ESI): calcd. for $C_{20}H_{18}N_2O_3S$ [M+Na]$^+$ 389.0930, found 389.0931.

## 4-Methyl-3-((4-(2-methylthiazol-4-yl)benzoyl)amino)benzamide (5)

Methyl 4-methyl-3-((4-(2-methylthiazol-4-yl)benzoyl)amino)benzoate

To 4-(2-methylthiazol-4-yl)benzoic acid (800 mg, 3.65 mmol) and methyl 3-amino-4-methylbenzoate (52 μL, 9.12 mmol) in tetrahydrofuran (15 mL) was added phosphorus trichloride (0.32 mL, 3.65 mmol). The reaction mixture was heated in a microwave reactor for 20 min at 150°C. The crude product was taken up in saturated aq. $NaHCO_3$ (5 mL) and extracted with ethyl acetate (3 × 10 mL). The combined organic components were then washed with brine (10 mL), dried over $MgSO_4$, filtered and concentrated under reduced pressure. The crude was purified by column chromatography (amino silica 12 g, 0% to 50% ethyl acetate in petroleum ether) to yield the desired amide as a white solid (920 mg, 65%). Rf 0.23 (petroleum ether:ethyl acetate 3:1); m.p. 179–181°C; [1]H-NMR (500 MHz, DMSO-d6) δ 10.03 (s, 1H, CONH), 8.14–8.07 (m, 3H, H-3, H-6), 8.05 (d, J 8.3, 2H, H-2), 8.00 (d, J 1.8, 1H, H-6'), 7.76 (dd, J 7.9, 1.8, 1H, H-4'), 7.44 (d, J 7.9, 1H, H-9'), 3.85 (s, 3H, OCH$_3$), 2.74 (s, 3H, 8-CH$_3$), 2.33 (s, 3H, 2'-CH$_3$); [13]C-NMR (126 MHz, DMSO-d6) δ 166.4 (CO), 165.6 (C-8), 153.3 (C-5), 140.0 (C-8'), 137.6 (C-1), 137.3 (C-7'), 133.7 (C-4), 131.3 (C-3'), 128.8 (C-2), 128.1 (C-6'), 127.5 (C-5'), 126.9 (C-4'), 126.3 (C-3), 116.1 (C-6), 52.5 (OCH$_3$), 19.4 (8-CH$_3$), 18.6 (2'-CH$_3$). COO not visible; IR (neat, $v_{max}$, cm$^{-1}$) 3256, 1726, 1643, 1523, 1432, 1296, 1171, 1110; LCMS (LCQ) Rt = 2.6 min (method 1), m/z (ESI+) 367.0 [M+H]$^+$; HRMS m/z (ESI): calcd. for $C_{20}H_{18}N_2O_3S$ [M+Na]$^+$ 389.0930, found 389.0931.

## 4-Methyl-3-((4-(2-methylthiazol-4-yl)benzoyl)amino)benzoic acid

To methyl 4-methyl-3-((4-(2-methylthiazol-4-yl)benzoyl)amino)benzoate (920 mg, 2.51 mmol) in methanol (24 mL) was added 2 M aq. sodium hydroxide (12.6 mL, 25.1 mmol) and the mixture stirred at an ambient temperature for 16 hr. The reaction mixture was then heated for 2 hr at 50°C. The solvent was removed under reduced pressure. The crude was taken up in water and the mixture was acidified to a pH of 2–3 with 2 M aq. hydrochloric acid solution. The resulting precipitate was collected by vacuum filtration and dried under reduced pressure to afford the desired acid as a white solid (926 mg, 99%). m.p. 254–256°C; [1]H-NMR (500 MHz, DMSO-d6) δ 12.85 (s, 1H, COOH), 10.04 (s, 1H, CONH), 8.13–8.07 (m, 3H, H-3, H-6), 8.06 (d, J 8.3, 2H, H-2), 7.95 (d, J 1.7, 1H, H-6'), 7.74 (dd, J 7.9, 1.7, 1H, H-4'), 7.40 (d, J 7.9, 1H, H-3'), 2.74 (s, 3H, 8-CH$_3$), 2.32 (s, 3H, 2'-CH$_3$); [13]C-NMR (126 MHz, DMSO-d6) δ 167.4 (COOH), 166.4 (CONH), 165.6 (C-8), 153.3 (C-5), 139.5 (C-2'), 137.5 (C-1), 137.1 (C-1'), 133.8 (C-4), 131.1 (C-3'), 129.3 (C-5'), 128.7 (C-2), 127.8 (C-6'), 127.2 (C-4'), 126.3 (C-3), 116.01 (C-6), 19.4 (8-CH$_3$), 18.59 (2'-CH$_3$); IR (neat, ν$_{max}$, cm$^{-1}$) 2923, 1679, 1638, 1512, 1492, 1414, 1390, 1251, 1178; LCMS (LCQ) Rt = 3.0 min (method 1), m/z (ESI+) 353.0 [M+H]$^+$; HRMS (ESI): calcd. for C$_{19}$H$_{16}$N$_2$NaO$_3$S [M+Na]$^+$ 375.0774, found 375.0774.

## 4-Methyl-3-((4-(2-methylthiazol-4-yl)benzoyl)amino)benzamide (5)

To 4- methyl-3-((4-(2-methylthiazol-4-yl)benzoyl)amino)benzoic acid (146) (100 mg, 0.28 mmol), EDCI (68 mg, 0.35 mmol), HOBt (54 mg, 0.35 mmol), N,N-diisopropylethylamine (100 μL, 0.57 mmol) in N, N-dimethylformamide (1 mL) was added 2 M ammonia in methanol (0.43 mL, 0.85 mmol). The reaction mixture was stirred at ambient temperature for 16 hr. The solvent was removed under reduced pressure. The crude product was taken up in ethyl acetate (5 mL), washed with 1 M aq. hydrochloric acid solution. (4 mL), saturated aq. NaHCO$_3$ (4 mL), brine (4 mL), dried over MgSO$_4$, filtered and concentrated under reduced pressure. The crude was purified by column chromatography (amino silica 12 g, 0% to 5% methanol in dichloromethane) to yield the desired primary amide five as a white solid (39 mg, 37%). Rf 0.17 (dichloromethane:methanol 19:1); m.p. 215–217°C; [1]H-NMR (500 MHz, DMSO-d6) δ 10.02 (s, 1H, CONH), 8.14–8.08 (m, 3H, H-3. H-6), 8.05 (d, J 8.5, 2H, H-2), 7.91 (s, 1H, 5'-CONHAB), 7.86 (d, J 1.8, 1H, H-6'), 7.70 (dd, J 7.9, 1.8, 1H, H-4'), 7.35 (d, J 7.9, 1H, H-3'), 7.27 (s, 1H, 5'-CONHAB), 2.74 (s, 3H, 8-CH$_3$), 2.28 (s, 3H, 2'-CH$_3$); [13]C-NMR (126 MHz, DMSO-d6) δ 167.9 (CO), 166.4 (CO), 165.5 (C-8), 153.3 (C-5), 138.0 (C-2'), 137.5 (C-1), 136.8 (C-1'), 133.8 (C-4), 132.8 (C-5'), 130.6 (C-3'), 128.7 (C-2), 126.6 (C-6'), 126.3 (C-3), 125.6 (C-4), 116.0 (C-6), 19.4 (8-CH$_3$), 18.37 (2'-CH$_3$); IR (neat, ν$_{max}$, cm$^{-1}$) 3108, 1713, 1668, 1608, 1571, 1501, 1436, 1410, 1278; LCMS (LCQ) Rt = 1.7 min (method 1), m/z (ESI+) 352.0 [M+H]$^+$; HRMS (ESI): calcd. for C$_{19}$H$_{17}$N$_3$NaO$_2$S [M+Na]$^+$ 374.0934, found 374.0933.

## 4-Methyl-3-((4-(2-methyloxazol-4-yl)benzoyl)amino)benzamide (6)

### Methyl 4-(2-methyloxazol-4-yl)benzoate

Methyl 4-(2-bromoacetyl)benzoate (400 mg, 1.56 mmol) was stirred in neat acetamide (276 mg, 4.67 mmol) at 160°C for 2 hr. Water was added to the reaction mixture. The resulting precipitate was collected by vacuum filtration and dried under reduced pressure to afford the desired compound 191 as a brown solid (308 mg, 87%) which was carried forward without further purification. [1]H-NMR (500 MHz, DMSO-d6) δ 8.07 (s, 1H, H-6), 8.00 (d, J 8.2, 2H, H-3), 7.89 (d, J 8.1, 2H, H-2), 3.86 (s, 3H, OCH$_3$), 2.48 (s, 3H, 8-CH$_3$).

### 4-(2-Methyloxazol-4-yl)benzoic acid

To methyl 4-(2-methyloxazol-4-yl)benzoate (191) (279 mg, 1.2 mmol) in methanol (5 mL) and water (1.5 mL) was added sodium hydroxide (150 mg, 3.86 mmol) and the reaction mixture stirred at ambient temperature for 16 hr. Upon completion, the reaction mixture was acidified to pH of 2–3 using 2 M aq. hydrochloric acid solution. The resulting precipitate was collected by vacuum filtration and dried under reduced pressure to afford the desired acid as a as a white solid (130 mg, 47%). [1]H-NMR (500 MHz, DMSO-d6) δ 8.56 (s, 1H, 11, H-6), 7.97 (d, J 8.0, 2H, H-3), 7.85 (d, J 8.0, 2H, H-2), 2.47 (s, 3H, 8- CH$_3$); LCMS (LCQ) Rt = 0.5 min (method 1), m/z (ESI+) 204.2 [M+H]+.

## 4-Methyl-3-((4-(2-methyloxazol-4-yl)benzoyl)amino)benzamide (6)

To 4-(2-methylthiazol-4-yl)benzoic acid (110 mg, 0.54 mmol) in dichloromethane (3 mL) was added oxalyl chloride (0.05 mL, 0.65 mmol) in a dropwise manner followed by $N,N$-dimethylformamide (10 μL). The reaction mixture was stirred at ambient temperature for 16 hr. The crude acyl chloride was then added to a stirred mixture of 3-amino-4-methylbenzamide (98 mg, 0.65 mmol) followed by the addition of $N,N$-diisopropylethylamine (0.47 mL, 2.71 mmol). The reaction mixture was stirred at ambient temperature for 2–16 hr. The solvent was removed under reduced pressure. The crude product was taken up in saturated aq. $NaHCO_3$ (5 mL) and extracted with ethyl acetate (3 × 10 mL). The combined organic components were then washed with brine (10 mL), dried over $MgSO_4$, filtered and concentrated under reduced pressure. The crude was purified by column chromatography (silica 12 g, 0% to 5% methanol in dichloromethane) to yield the desired amide 6 as a white solid (55 mg, 29%). Rf 0.28 (dichloromethane:methanol 19:1); m.p. 260–262°C; $^1$H-NMR (500 MHz, DMSO-d6) δ 10.00 (s, 1H, CONH), 8.61 (s, 1H, H-6), 8.05 (d, J 8.1, 2H, H-3), 7.91 (d, J 8.4, 3H, H-2, 5'-CONHAB), 7.86 (s, 1H, H-6'), 7.70 (dd, J 7.9, 1.9, 1H, H-4'), 7.35 (d, J 7.9, 1H, H-3'), 7.27 (s, 1H, 5'-CONHAB) 2.28 (s, 3H, 2'-CH3). 8-CH3 under DMSO peak; $^{13}$C-NMR (126 MHz, DMSO-d6) δ 167.8 (CO), 165.4 (CO), 162.2 (C-8), 139.4 (C-5), 137.9 (C-2'), 136.8 (ArC), 136.3 (ArC), 134.6 (C-1), 133.8 (C-4), 132.8 (C-5'), 130.6 (C-3'), 128.7 (C-2), 126.6 (C-6'), 125.5 (C-4'), 125.33 (C-3), 18.4 (8-CH3), 14.0 (2'-CH3); IR (neat,, $v_{max}$, $cm^{-1}$) 3255, 1673, 1635, 1616, 1522, 1489, 1386, 1276, 1214; LCMS (LCQ) Rt = 0.8 min (method 1), m/z (ESI+) 336.1 [M+H]$^+$; HRMS (ESI): calcd. for $C_{19}H_{17}N_3NaO_3$ [M+Na]$^+$ 358.1162, found 358.1150.

## *N*-(6-methyl-2-pyridyl)-4-(2-methylthiazol-4-yl)benzamide (7)

To 4-(2-methylthiazol-4-yl)benzoic acid (80 mg, 0.36 mmol) in dichloromethane (3 mL) was added oxalyl chloride (37 μL, 0.44 mmol) in a dropwise manner followed by $N,N$-dimethylformamide (10 μL). The reaction mixture was stirred at ambient temperature for 16 hr. The crude acyl chloride was then added to a stirred mixture of 2-amino-6-methylpyridine (40 mg, 0.36 mmol) followed by the addition of $N,N$-diisopropylethylamine (397 μL, 2.28 mmol). The reaction mixture was stirred at ambient temperature for 2–16 hr. The solvent was removed under reduced pressure. The crude product was taken up in saturated aq. $NaHCO_3$ (5 mL) and extracted with ethyl acetate (3 × 10 mL). The combined organic components were then washed with brine (10 mL), dried over $MgSO_4$, filtered and concentrated under reduced pressure. The crude was purified by column chromatography (silica 12 g, 0% to 45% ethyl acetate in petroleum ether) to yield the desired amide 7 as a colourless solid (54 mg, 45%). Rf 0.48 (petroleum ether:ethyl acetate 1:1); m.p. 195–197°C; $^1$H-NMR (500 MHz, DMSO-d6) 10.69 (s, 1H), 8.13 (s, 1H), 8.10 (d, J 8.2, 2H), 8.06 (d, J 8.0, 2H), 8.02 (d, J 8.2, 1H), 7.73 (t, J 7.8, 1H), 7.03 (d, J 7.3, 1H), 2.74 (s, 3H), 2.46 (s, 3H); $^{13}$C-NMR (126 MHz, DMSO-d6) 166.4, 165.9, 157.0, 153.3, 152.0, 138.8, 137.6, 133.5, 129.1, 126.1, 119.5, 116.2, 112.1, 24.0, 19.4; IR (neat, $v_{max}$, $cm^{-1}$) 3314, 2923, 1610, 1539, 1522, 1290, 1169; LCMS (LCQ) Rt = 3.1 min (method 1), m/z (ESI+) 310.1 [M+H]$^+$; HRMS (ESI): calcd. for $C_{17}H_{16}N_3OS$ [M+H]$^+$ 310.1009, found 310.1012.

## 1-(4-Fluoro-3-(trifluoromethyl)phenyl)-3-(5-(4-pyridyl)-1,3,4-thiadiazol-2-yl)urea (ML216) (*Rosenthal et al., 2013*)

Phenyl-5-(4-pyridyl)−1,3,4-thiadiazol-2-ylcarbamate

Sodium hydride (700 mg, 33.7 mmol) was slowly added to a suspension of 5-(4-pyridyl)−1,3,4-thia-diazol-2-yl-amine (2.00 g, 11.2 mmol) in tetrahydrofuran (40 mL) at 0°C. The resulting reaction mixture was stirred at 0°C for 2 hr. Diphenyl carbonate (2.89 g, 13.5 mmol) was added and the reaction mixture was stirred at 0°C for 30 min. The reaction mixture was warmed to ambient temperature and stirred overnight. Dichloromethane (40 mL) and brine (10 mL) was added to the reaction mixture and the solid precipitate was collected by filtration to yield the desired compound as a crystalline off-white solid (3.10 g, 93%). Rf 0.52 (dichloromethane:methanol 19:1); m.p. 278–280°C; $^1$H-NMR (500 MHz, DMSO-d6) δ 8.58 (d, J 4.9, 2H, H-2), 7.70 (d, J 4.9, 2H, H-3), 7.33 (t, J 7.6, 2H, H-15), 7.11 (t, J 7.4, 1H, H-17), 7.07 (d, J 7.7, 2H, H-16); $^{13}$C-NMR (126 MHz DMSO-d6) δ = 174.7 (CO), 162.1 (ArC), 155.1 (ArC), 153.7 (C-14), 150.7 (C-2), 140.1 (ArC), 129.2 (C-15), 124.0 (C-17), 122.4 (C-16), 120.2 (C-3); IR (neat, $v_{max}$, $cm^{-1}$) 3542, 3118, 2417, 1604, 1460, 1319, 1296, 1208, 1113; LCMS

(LCQ) Rt = 1.8 min (method 2), m/z (ESI+) 299.03 $[M+H]^+$; HRMS (ESI): calcd. for $C_{14}H_{11}N_4O_2S$ $[M+H]^+$ 299.0598, found 299.0597.

## 1-(4-Fluoro-3-(trifluoromethyl)phenyl)−3-(5-(4-pyridyl)−1,3,4-thiadiazol-2-yl) urea (ML216)

To a suspension of phenyl -(5-(4-pyridyl)−1,3,4-thiadiazol-2-yl)carbamate (150 mg, 0.50 mmol) in toluene (5 mL) was added 4-fluoro-3-(trifluoromethyl)aniline (65 µL, 0.50 mmol). The reaction mixture was heated in a microwave reactor at 150°C for 30 min and the resulting suspension was cooled to room temperature. The reaction mixture was concentrated under reduced pressure. The resulting solid 223 was triturated with dichloromethane (5 mL) and further triturated with 5% methanol in dichloromethane (3 mL) to yield the desired compound ML216 as an orange solid (82 mg, 46%). $^1$H-NMR (500 MHz, DMSO-d6) δ 11.61 (s, 1H, NH-urea), 9.52 (s, 1H, NH-urea), 8.73 (d, J 5.9, 2H, H-2), 8.06 (s, 1H, H15), 7.89 (d, J 5.3, 2H, H-3), 7.77 (s, 1H, H-19), 7.49 (t, J 9.7, 1H, H-18); LCMS (LCQ) Rt = 2.7 min (method 1), m/z (ESI+) 384.0 [M+H]+. $^1$H-NMR consistent with literature data (*Rosenthal et al., 2013*).

**Appendix 1—table 1.** Statistics for data collection, phasing and refinement.

| | BLM-HD$^{\Delta WHD}$ + ADP | Liganded-BLM-HD$^{\Delta WHD}$ |
|---|---|---|
| Data collection | | |
| Space group | P2$_1$ | P1 |
| Cell dimensions | | |
| a, b, c (Å) | 54.28, 107.69, 55.20 | 84.69, 111.60, 132.38 |
| α, β, γ (°) | 90.00, 109.31, 90.00 | 72.70, 80.13, 79.24 |
| Wavelength | 0.9780 | 0.9762 |
| Resolution (Å) | 51.23–1.53 (1.56–1.53) | 125.37–2.97 (3.08–2.97) |
| Mn I / σI | 12.7 (1.2) | 7.8 (1.4) |
| Mn I, CC$_{1/2}$ | 1.00 (0.61) | 0.99 (0.57) |
| Completeness (%) | 98.3 (90.3) | 98.0 (94.5) |
| Redundancy | 1.9 (1.7) | 2.6 (2.7) |
| | | |
| Refinement | | |
| Resolution (Å) | 51.23–1.53 (1.56–1.53) | 47.28–2.97 (3.07–2.96) |
| No. unique reflections | 88464 (8096) | 91661 (8892) |
| R$_{work}$ / R$_{free}$ | 0.19/0.21 | 0.23/0.27 |
| No. atoms | | |
| Macromolecules | 3870 | 24954 |
| Ligands | 65 | 427 |
| Solvent | 438 | 91 |
| B-factors | | |
| Wilson | 32.58 | 77.77 |
| ADP (mean) | | |
| Macromolecules | 31.00 | 95.79 |
| Ligands | 40.89 | 98.56 |
| Solvent | 45.69 | 53.65 |
| R.m.s. deviations | | |
| Bond lengths (Å) | 0.014 | 0.006 |
| Bond angles (°) | 1.64 | 1.11 |

*Continued on next page*

*Appendix 1—table 1 continued*

|  | BLM-HD$^{\Delta WHD}$ + ADP | Liganded-BLM-HD$^{\Delta WHD}$ |
|---|---|---|
| Molprobity |  |  |
| All atom clashscore | 2.44 | 7.42 |
| Ramachandran |  |  |
| Outliers | 0.21% | 0.42% |
| Allowed | 1.65% | 3.72% |
| Favoured | 98.15% | 95.85% |

*Values in parentheses are for the highest resolution shell

