## [Decision Letter]

**Acceptance summary:**

The RecQ helicase family of genome maintenance proteins play crucial roles in the DNA damage response. In this work, Chen et al., identify a new selective compound that inhibits the unwinding activity of the BLM (Bloom Syndrome) helicase. The compound appears to be an interesting first step toward developing a cellular BLM inhibitor. The authors also find that the HRDC domain of the BLM helicase, in complex with the compound, interacts with the ssDNA. This provides insight into the mechanism and function of this domain in mediating helicase activity.

**Decision letter after peer review:**

Thank you for submitting your article "Uncovering an allosteric mode of action for a selective inhibitor of human Bloom syndrome protein" for consideration by *eLife*. Your article has been reviewed by three peer reviewers, one of whom is a member of our Board of Reviewing Editors, and the evaluation has been overseen by Cynthia Wolberger as the Senior Editor. The reviewers have opted to remain anonymous.

The reviewers have discussed the reviews with one another and the Reviewing Editor has drafted this decision to help you prepare a revised submission.

Summary:

The RecQ helicase family of genome maintenance proteins play crucial roles in the DNA damage response. In this work, Chen et al., identify a new selective compound that inhibits the unwinding activity of the BLM (Bloom Syndrome) helicase. Crystallographic studies reveal that this compound locates at an unknown interdomain interface that is postulated to interfere with the conformational change of the helicase during unwinding but not DNA binding. The compound appears to be an interesting first step toward developing a cellular BLM inhibitor. It has an IC50 of 1.8uM and although is poorly cell permeable will make a good starting point for future studies.

The authors also find that the HRDC domain of the BLM helicase, in complex with the compound, interacts with the ssDNA. This provides insight into the mechanism and function of this domain in mediating helicase activity.

Essential revisions:

1) As a general reader I found the discussion of the BLM mechanism and the role of the HRDC domain difficult to follow. Could the authors include a bit more background and outline the key questions that are unclear about the current proposed mechanism? It would also help if the authors started the relevant Discussion section with a summary of what the insights are, then follow it with the detailed explanation for how they came to the conclusion. At the moment the sections tend to dive straight into detail first.

2) The authors should be more cautious when discussing the HRDC domain. The crystal structure of BLM-DNA-ADP-inhibitor complexes was obtained by using an artificial DNA substrate with partial complementary between two ssDNA and a truncated protein. The HRDC domain is known to associate with ssDNA, albeit at low efficiency. With their DNA substrate, it is not surprising that the HRDC interacts with a nearby ssDNA. In terms of forked DNA and Holliday junction substrates, the HRDC domain may be positioned in other places or maybe very dynamic when additional dsDNA is provided.

3) The conclusion that the compound traps the BLM helicase in the "closed" state is too speculative. The notion of the "open" and "closed" states of BLM was derived from a kinetic study where a structural transition between two ADP-bound states was postulated. Finding that BLM complexed with the compound could not unwind DNA is not sufficient to draw the aforementioned conclusion.

4) As presented, it is unclear from the text what genetic contexts that BLM inactivation could be exploited in a synthetic lethal approach? Are there clear applications for BLM inhibitors?

5) Does compound 2 binding influence conformational changes proximal the HRDC docking site on D1 and D2, making it incompatible with HRDC domain docking? That is, could the inhibitor also be influencing conformational switching of the HRDC domain from the parked to engaged states?

Experimental suggestions – optional :

1) Does the crystallized WHD deletion construct have helicase activity or DNA-stimulated ATPase activity? The report only assesses DNA binding for the construct used for structural work. Are the helicase or ATPase activities inhibited by compound 2 for the WHD deletion construct? If so, is the inhibitor more or less potent for the WHD deletion construct. Given the WHD domain's proximity to the D2 and Zn-binding domains, its deletion may influence compound efficacy. This could be tested directly with reagents in hand, or this caveat could be raised in the Discussion as WHD deletion may very well be impacting BLM structural dynamics and/or the states observed in the inhibitor bound crystal structure of the WHD deletion protein.

2) The inhibitory effect of the compound was examined with BLM-HD only. Additional experiments with full-length BLM would make the effect more convincing. Besides, BLM is known to be able to unwind a collection of DNA substrates, such as G-quadruplex and Holliday Junction. It will further generalize the conclusion by testing if the compound also inhibits BLM-mediated unwinding of other DNA structures besides forked DNA.

3) An EMSA assay would be more direct to show that BLM can still bind to the DNA substrate in the presence of compound 2, whereas ML216 affects the DNA binding by BLM.

---

## [Author Response]

Essential revisions:1) As a general reader I found the discussion of the BLM mechanism and the role of the HRDC domain difficult to follow. Could the authors include a bit more background and outline the key questions that are unclear about the current proposed mechanism? It would also help if the authors started the relevant Discussion section with a summary of what the insights are, then follow it with the detailed explanation for how they came to the conclusion. At the moment the sections tend to dive straight into detail first.

We have tried as much as possible to make the manuscript follow a logical narrative, which leads on from the discovery that compound **2** is an allosteric inhibitor of BLM, to a set of observations that can be made from analysing the resultant structure in more detail. However, we appreciate that in parts the document is quite technical in detail and have now included an additional paragraph at the start of the Discussion to hopefully aid readers who may have less background knowledge around the subject area.

2) The authors should be more cautious when discussing the HRDC domain. The crystal structure of BLM-DNA-ADP-inhibitor complexes was obtained by using an artificial DNA substrate with partial complementary between two ssDNA and a truncated protein. The HRDC domain is known to associate with ssDNA, albeit at low efficiency. With their DNA substrate, it is not surprising that the HRDC interacts with a nearby ssDNA. In terms of forked DNA and Holliday junction substrates, the HRDC domain may be positioned in other places or maybe very dynamic when additional dsDNA is provided.

We have placed our thoughts and discussion about the HRDC domain into a subsection of the Discussion very carefully titled “Speculative model for the involvement of the HRDC in unwinding DNA substrates”. At each step, we also have taken care to compare our proposed model back to the scientific literature and to indicate where it might help to provide a rational explanation for each of its known cellular roles.

We didn’t in fact foresee the self-interaction of the single-stranded DNA oligonucleotide, although this does become evident with hindsight. As far as we know, no other structure of a RecQ-helicase has captured the interaction of the HRDC domain with ssDNA, with all previous structures of BLM-HD in complex with ssDNA having the HRDC bound at the “parked” position.

We believe that we have able to capture this v. weak interaction, as a downstream consequence of the inhibitor stabilising the interaction between the oligonucleotide and the D1 domain. We also agree with the reviewer that the HRDC may be positioned in other places when other types of DNA structure/substrate are encountered; we chose to provide a model for what is believed to be the preferred substrate of BLM.

3) The conclusion that the compound traps the BLM helicase in the "closed" state is too speculative. The notion of the "open" and "closed" states of BLM was derived from a kinetic study where a structural transition between two ADP-bound states was postulated. Finding that BLM complexed with the compound could not unwind DNA is not sufficient to draw the aforementioned conclusion.

In light of this comment, we have taken the opportunity to revisit and revise this particular section of our Discussion. We do however provide a set of MST data that indicate that trapping does in fact occur in vitro (please see Figure 6B).

4) As presented, it is unclear from the text what genetic contexts that BLM inactivation could be exploited in a synthetic lethal approach? Are there clear applications for BLM inhibitors?

We have provided several citations (within the Introduction) to manuscripts that explore more fully the potential of BLM (and WRN) as therapeutic targets. However, we have now added an additional short statement at the end of the manuscript to hopefully address this comment, including a timely review exploring synthetic lethal interactions of RECQ helicases, that has just become available in Trends in Cancer (Datta et al., 2020). Ultimately, generation of a cell-penetrant tool compound that selectively inhibits BLM will allow these synthetic lethal/sick relationships to be explored across a range of tumour backgrounds.

5) Does compound 2 binding influence conformational changes proximal the HRDC docking site on D1 and D2, making it incompatible with HRDC domain docking? That is, could the inhibitor also be influencing conformational switching of the HRDC domain from the parked to engaged states?

The reviewer makes a very interesting point here, which has prompted us to revisit and revise the Discussion section of our manuscript.

Experimental suggestions – optional:1) Does the crystallized WHD deletion construct have helicase activity or DNA-stimulated ATPase activity? The report only assesses DNA binding for the construct used for structural work. Are the helicase or ATPase activities inhibited by compound 2 for the WHD deletion construct? If so, is the inhibitor more or less potent for the WHD deletion construct. Given the WHD domain's proximity to the D2 and Zn-binding domains, its deletion may influence compound efficacy. This could be tested directly with reagents in hand, or this caveat could be raised in the Discussion as WHD deletion may very well be impacting BLM structural dynamics and/or the states observed in the inhibitor bound crystal structure of the WHD deletion protein.

We can confirm that BLM-HD^DWHD^ has DNA-stimulated ATPase activity, but does not have unwinding activity; consistent with a requirement for the deleted WHD domain to bind and stabilise the double-stranded DNA section of an incoming substrate. So, for consistency across our assay formats, we chose to use BLM-HD in preference to BLM-HD^DWHD^, where possible.

We would like to point the reviewer at Figure 2F, which shows binding of BLM-HD to ssDNA, and to Figure 3—figure supplement 2 which shows binding of BLM-HD^DWHD^ to the same 15mer. Likewise, we refer the reviewer to Figure 2G which shows binding of compound 2 to BLM-HD and to Figure 3—figure supplement 2 that shows binding of 2 to BLM-HD^DWHD^ (both in the context of a preformed complex with ssDNA). We also make the following statement in the manuscript itself, “… BLM-HD^DWHD^ binds both ssDNA-15mer and **2** with a similar affinity to that of BLM-HD”.

2) The inhibitory effect of the compound was examined with BLM-HD only. Additional experiments with full-length BLM would make the effect more convincing. Besides, BLM is known to be able to unwind a collection of DNA substrates, such as G-quadruplex and Holliday Junction. It will further generalize the conclusion by testing if the compound also inhibits BLM-mediated unwinding of other DNA structures besides forked DNA.

Unfortunately, we have found full-length BLM difficult to make in the laboratory. Whilst we can detect expression of the protein in insect cells by western blot, we are as yet unable to make sufficient quantities at a purity that would be suitable for our biochemical assays. This is definitely an experiment on our “to-do” list. However, as a consequence of the current COVID-19 situation in the UK, access to the laboratory is heavily time-restricted, and so this is not something we would be able to readily able to achieve in a timely fashion.

It is worth noting that all unwinding activities of the RecQ helicase family depend on the translocation of single-stranded DNA – from making interactions with the D2 domain to those with the D1 domain in an “inchworm” type of mechanism. Therefore, if a compound prevents this from happening, then unwinding cannot occur, no matter the type of substrate used.

3) An EMSA assay would be more direct to show that BLM can still bind to the DNA substrate in the presence of compound 2, whereas ML216 affects the DNA binding by BLM.

We agree with the reviewer that this type of experiment could be of use. However, we have not optimised this particular method using BLM-HD. We believe that our MST, FP and indeed crystallographic data are sufficient to demonstrate that BLM can bind to a DNA substrate in the presence of compound **2**. We also refer the reviewer to Nygen et al., 2013 – where ML216 has previously been shown to disrupt the ability of BLM to bind to both ssDNA and a forked DNA duplex by EMSA.